# Significant stability enhancement in photocatalytic CO₂ reduction via flow-driven strategies

Hyunju Jung [1,2,3], Hyo Sang Jeon [4], Min Gyu Kim [5], Aqil Jamal [6], Issam Gereige[6], Chansol Kim [7] ✉ & Hee-Tae Jung [1,2,3] ✉

Achieving long-term stability remains a major challenge in photocatalytic $CO_2$ reduction. Unlike natural photosynthesis, most artificial systems exhibit severe activity losses within hours due to catalyst deactivation and surface degradation. This study investigates the effect of continuous $CO_2$ and $H_2O$ flow during the photocatalytic process. Under optimized flow conditions, widely used photocatalysts such as $TiO_2$, ZnO, CdS, and $C_3N_4$ show up to 50-fold improvement in operational stability, with $TiO_2$ retaining 80% of its initial activity over 15 days. $CO_2$ flow plays a more dominant role than $H_2O$ flow, mitigating product accumulation and preventing catalyst deactivation. Surface and structural analyses reveal that systems without flows suffer from product and intermediate accumulation, while flow-enabled systems maintain clean catalytic surfaces. X-ray absorption spectroscopy confirms the suppression of structural degradation under flow. Here, we establish flow control as a design principle for durable photocatalytic $CO_2$ reduction, providing a pathway for scalable solar-to-chemical energy conversion.

The photocatalytic reduction of $CO_2$ into value-added chemicals such as carbon monoxide (CO), methane ($CH_4$), and methanol ($CH_3OH$)[1–3] offers a promising route for sustainable solar-to-chemical energy conversion, inspired by the high efficiency and longevity of natural photosynthesis[4–10]. This process not only addresses the pressing issue of rising atmospheric $CO_2$ levels but also provides a pathway to renewable fuel production. Despite extensive research efforts, a significant hurdle remains: the rapid deactivation of photocatalysts under operational conditions[11,12]. Typically, these system experience over 80% loss of activity within h[13], primarily due to catalyst degradation mechanisms such as photo-corrosion[14,15], surface oxidation[16,17], and the accumulation of reaction intermediates or carbonaceous residues that block active sites[18]. This instability clearly contrasts with the robustness of natural photosynthetic systems, which operate efficiently over extended periods.

Previous strategies to enhance photocatalyst stability have largely focused on material modifications, including bandgap engineering[19,20], co-catalyst integration[21,22], and surface passivation[23,24]. While these approaches have led to modest stability improvements, typical operational durations still remain below ~10 h[25–27]. Furthermore, they usually involve complex, multistep syntheses that hinder scalability and reproducibility.

From a reactor system-engineering perspective rather than further materials modification, numerous flow-based photoreactors, including gas–solid, liquid–solid, and gas–liquid–solid configurations, have been reported to enhance illumination and mass transport[11,28–31]

¹Department of Chemical and Biomolecular Engineering, Korea Advanced Institute of Science and Technology (KAIST), Daejeon, South Korea. ²KAIST-UC Berkeley-Vietnam National University Climate Change Research Center, Daejeon, South Korea. ³Saudi Aramco-KAIST CO2 Management Center, Daejeon, South Korea. ⁴Sustainable Energy Research Division, Korea Institute of Science and Technology (KIST), Seoul, South Korea. ⁵Beamline Research Division, Pohang Accelerator Laboratory, Pohang University of Science and Technology, Pohang, South Korea. ⁶Research and Development Center, Saudi Aramco, Dhahran, Saudi Arabia. ⁷Clean Energy Research Center, Korea Institute of Science and Technology (KIST), Seoul, South Korea. ✉e-mail: chansolkim@kist.re.kr; heetae@kaist.ac.kr

(Supplementary Fig. 1a-b). Nevertheless, most of these studies emphasize performance enhancement, while the role of continuous flow in regulating catalyst deactivation and long-term stability has not been systematically investigated.

Recognizing these limitations, a continuous-flow three-phase (gas–liquid–solid) photocatalytic systems have recently been explored to improve reactant accessibility and mass transport[32] (Supplementary Fig. 1c) This system demonstrated significantly enhanced production rates and stability by maintaining simultaneous gas and liquid flow at opposite sides of the catalyst layer. However, the fundamental mechanisms underlying this considerable stability improvement remained unclear, particularly regarding which flow component is more critical and through what surface-level processes deactivation is suppressed.

In this study, we elucidate why the three-phase flow configuration achieves such enhanced stability. By comparing flow-enabled and flow-less conditions we reveal that this approach mitigates mass transport limitations, prevents the accumulation of products and intermediates, and inhibits the formation of deactivating surface species. Under optimized flow conditions, we demonstrate a significant enhancement in the long-term stability of widely used photocatalysts, including $TiO_2$, ZnO, CdS, and $C_3N_4$. Notably, $TiO_2$ maintained 80% of its initial activity over 15 days, representing a competitive operational stability among reported photocatalytic $CO_2$ reduction systems. Our findings suggest that incorporating continuous flow conditions can serve as a foundational design principle for developing highly durable photocatalytic $CO_2$ reduction systems.

## Results

### Flow-enabled photocatalytic system

To address the impact of continuous $CO_2$ and $H_2O$ flow on photocatalyst stability, we compared a flow-enabled photocatalytic system with a conventional flow-less batch system (Fig. 1a, b). In the absence of continuous $CO_2$ and $H_2O$ flow (Fig. 1a), mass transfer relies only on diffusion, leading to the accumulation of products and intermediates on the catalyst surface. For instance, in $TiO_2$-based systems, localized accumulation of CO near active sites can elevate the local partial pressure relative to the bulk environment. Such conditions can promote undesired side reactions, including re-oxidation of CO to $CO_2$, thereby reducing overall production efficiency[33,34]. Additionally, the buildup of products accelerates the attainment of reaction equilibrium, diminishing reaction rates and increasing the likelihood of further side reactions. Products that are not promptly desorbed may adhere to the catalyst surface, causing poisoning effects that reduce the effective catalytic surface area, leading to decreased initial activity and gradual deactivation[35,36].

Introducing a continuous flow of $CO_2$ and $H_2O$ across the photocatalyst surface addresses these limitations by enhancing mass transport (Fig. 1b and Supplementary Figs. 1, 2). Quantitative analysis revealed a 15-fold enhancement in $CO_2$ mass transfer in our flow-enabled system compared to the flow-less configuration, as validated through $CO_2$ uptake experiments under strong-sink conditions (Supplementary Fig. 3). This dynamic flow facilitates the efficient removal of products and intermediates from the catalyst surface, preventing their accumulation and associated deactivation effects. The continuous supply of reactants ensures a consistent reaction environment, reducing the formation of deactivating surface species and maintaining catalyst activity over extended periods. By comparing these two systems, the advantages of continuous flow in enhancing photocatalytic performance and stability become evident, underscoring its potential for sustainable solar-to-chemical energy conversion applications.

While the definition of photocatalyst stability varies across the literature[11], we define stability herein as the operational duration during which the product generation rate reaches 80% of its initial value. Mathematically, we define the Stability of Photocatalytic Reduction (SPR) as:

$$SPR = \text{Duration time where } \frac{p_t}{p_0} = 0.8 \qquad (1)$$

where $p_t$ is the production rate at time t, and $p_0$ is the initial production rate measured during the first hour of the reaction. For example, if the CO production rate of $TiO_2$ declines to 80% of its initial value after 1 h, we define the photocatalyst's stability as 1 h. This definition serves as a practical and standardized metric, enabling consistent comparisons across different studies.

Under this definition, the stability (SPR ≥ 0.8) of typical photocatalysts for $CO_2$ reduction is presented in Supplementary Fig. 4a, b. In traditional, flow-less batch systems—both gas-phase and liquid-phase—photocatalyst stability is severely constrained, often limited to 1–10 h, as shown in Supplementary Fig. 4b and Supplementary Table 1. In other words, most flow-less systems lose 80% of their catalytic activity within 1–10 h. This limitation is particularly evident in widely used photocatalysts such as $TiO_2$, ZnO, CdS, and $C_3N_4$. While minor improvements in stability have been reported through surface modifications and optimized co-catalysts, these approaches still fall short of practical, scalable solutions[11,37,38].

Notably, the flow-enabled platform demonstrated significantly enhanced stability. For instance, $TiO_2$ retained over 80% of its initial activity for more than 360 h (>15 days), achieving cumulative product formation rates up to 825 μmol·g⁻¹·h⁻¹—far exceeding the performance observed under batch conditions without $CO_2$ and $H_2O$ flow. This catalyst exhibits considerable stability, surpassing that of recently reported semiconductor-based $CO_2$ photocatalysts incorporating co-catalysts or heterostructures. The carbon source of CO

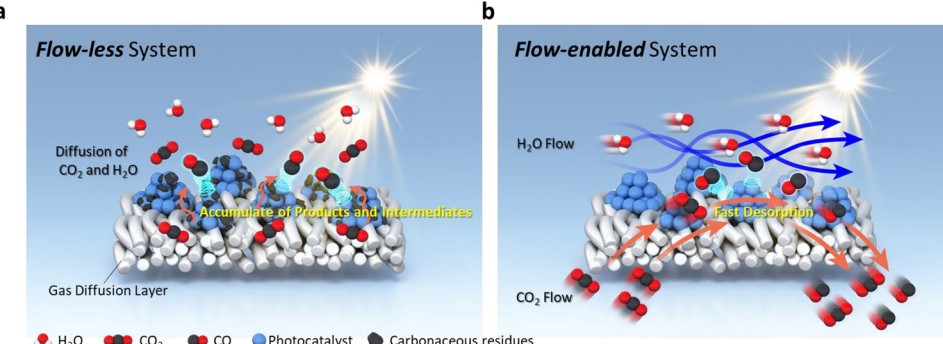

**Fig. 1 | Enhanced photocatalytic $CO_2$ reduction stability enabled by a continuous flow system. a** Schematic illustration of the flow-less batch system, showing inefficient mass transport and susceptibility to surface contamination. **b** Schematic illustration of the flow-enabled system, depicting improved mass transport and a stable reaction environment.

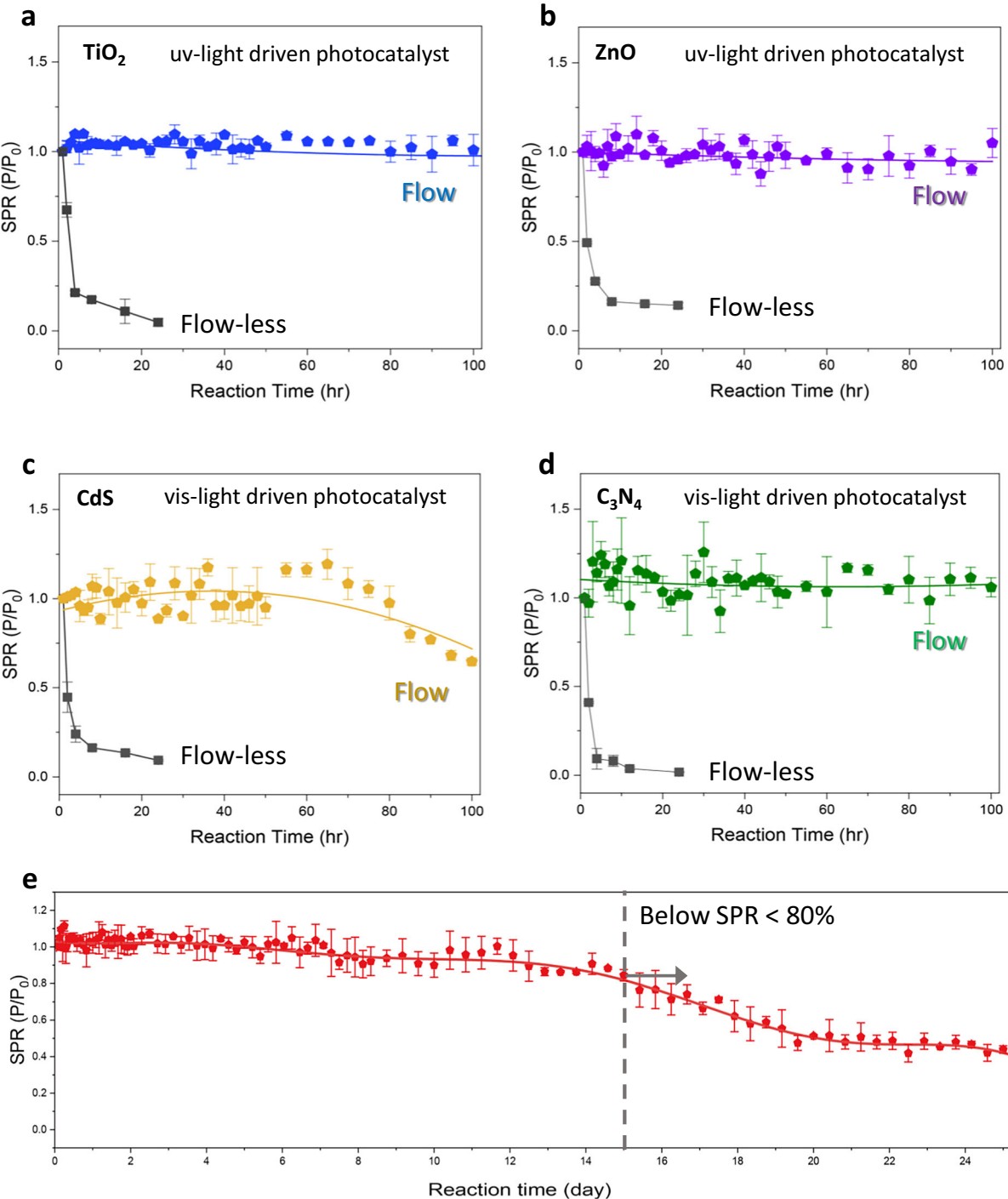

**Fig. 2 | Universality and extended long-term stability of the continuous flow-enabled photocatalytic CO₂ reduction system.** Stability performance of various photocatalysts, including **a** TiO₂, **b** ZnO, **c** C₃N₄, and **d** CdS over 100 h reaction times, demonstrating sustained photocatalytic activity under continuous flow and flow-less conditions. In each graph, the dark curve represents the performance of the corresponding photocatalyst under flowless batch conditions, with measurements recorded at 2, 4, 8, 18, and 24 h of reaction time. Both systems operated under identical pressure conditions, confirming that the enhanced stability arises from convective mass transport rather than pressure effects. **e** Extended long-term stability assessment of photocatalytic CO production over a 25-day period under continuous flow operation, showing a gradual decline in activity after 15 days. During the reaction period, measurements were conducted at 1-h intervals for the first 10 h, 2-h intervals up to 50 h, 5-h intervals up to 200 h (for **a**–**d**: up to 100 h), and 10-h intervals thereafter until the end of the experiment. All photocatalysts were tested under 300 mW/cm² illumination with an Xe lamp. Error bars represent the standard deviations from three replicate experiments.

production in this flow-enabled reactor configuration has been verified through ¹³CO₂ isotope labeling experiments in our previous work[32], confirming that CO originates from photocatalytic CO₂ reduction. This represents a competitive level of operational stability among previously reported TiO₂-based photocatalysts, and including recent systems employing co-catalysts or heterostructure engineering (Supplementary Fig. 5 and Supplementary Table 2), demonstrating the potential of flow-regulated operation in addressing longstanding barriers to practical photocatalytic CO₂ reduction.

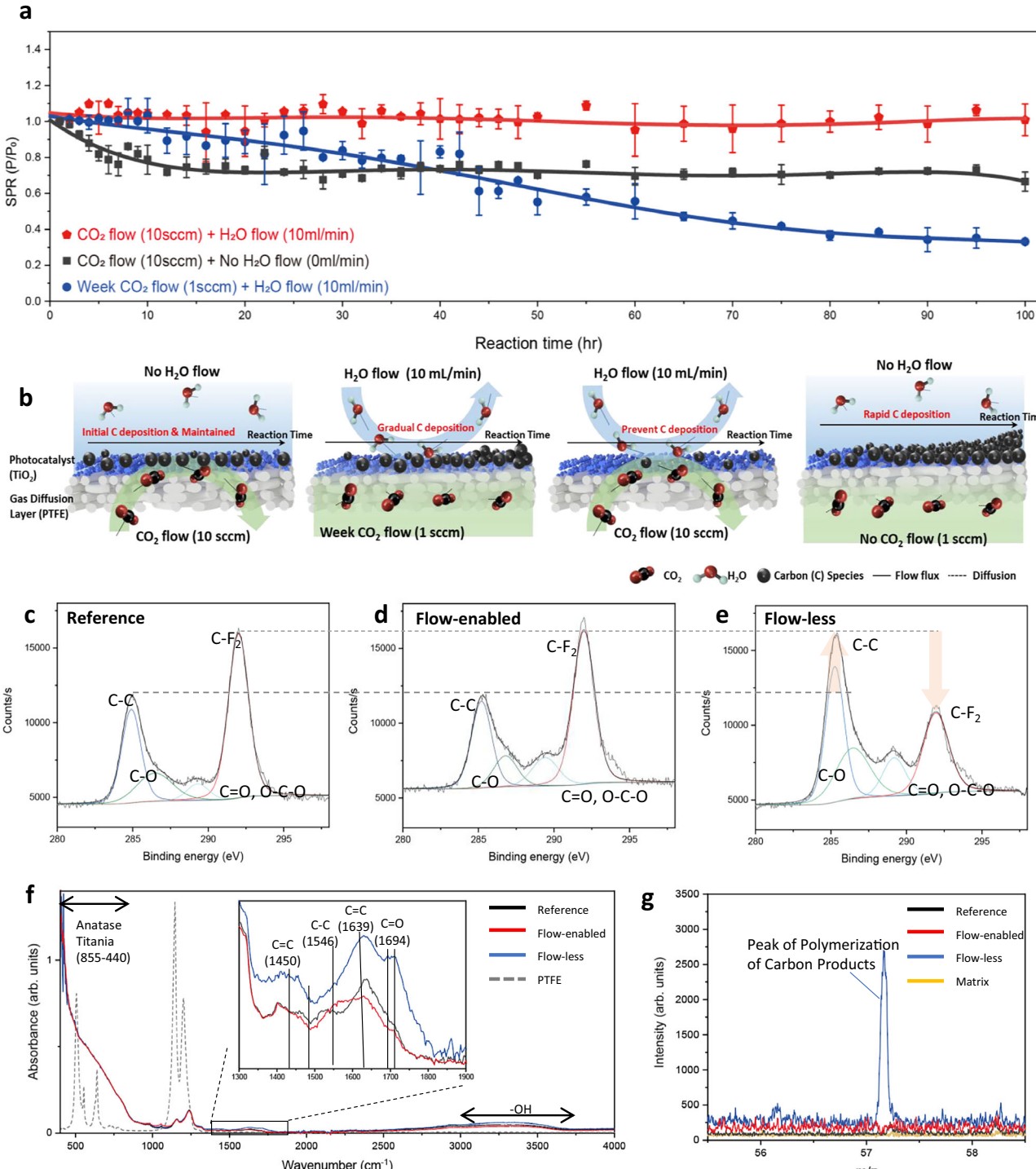

**Fig. 3 | Dominant factors in photocatalyst stability and deactivation mechanism revealed by various surface analysis under flow-enabled and flow-less system. a** Influence of gas and H₂O flow rates on photocatalyst stability, highlighting the optimal reaction conditions where both CO₂ flow (10 sccm) and H₂O flow (10 mL/min) are maintained. Reduced stability and performance were observed under suboptimal conditions, including the absence of H₂O flow or the presence of weak CO₂ flow. P25 TiO₂ photocatalyst was used under 300 mW·cm⁻² illumination with a Xe lamp. **b** Schematic illustration of different flow conditions applied to the photocatalyst layer: (from left to right) CO₂ flow (10 sccm) + no H₂O flow, weak CO₂ flow (1 sccm) + H₂O flow (10 mL·min⁻¹), CO₂ flow (10 sccm) + H₂O flow (10 mL·min⁻¹), and no CO₂ flow + no H₂O flow. **c**, XPS C *1s* spectrum of the

reference photocatalyst layer, **d** XPS C *1s* spectrum of the photocatalyst layer from the flow-enabled system after 100 h of reaction, and **e**, XPS C *1s* spectrum of the photocatalyst layer from the flow-less system after 100 h of reaction. **f** FTIR spectra of the reference catalyst and catalysts after 100 h of reaction under flow-enabled and flow-less batch conditions. The inset shows an enlarged view of the 1300–1900 cm⁻¹ region. **g**, MALDI-ToF mass spectra showing a distinct peak at 57 m/z in the flow-less batch sample, corresponding to high-carbon products such as butadiene, which is absent in the reference and flow-enabled samples. Error bars in **a** represent standard deviations from three replicate experiments. All photocatalysts were tested under 300 mW·cm⁻² illumination with a Xe lamp.

## System versatility and stability

Figure 2 illustrates the broad applicability and extended long-term stability of flow-enabled photocatalytic approach across various materials, including $TiO_2$, $ZnO$, $C_3N_4$, and $CdS$, over 100-h reaction times. Without any modifications to the photocatalysts, the $CO_2$ reduction activity of each material was evaluated under both continuous flow-enabled and traditional flow-less gas-phase batch system. The internal temperature of the reactor, measured via thermocouple insertion into the liquid plate, remained stable at ~32 °C throughout the 7-h operation under standard conditions (Supplementary Fig. 6).

UV-light-driven photocatalysts, such as $TiO_2$ and $ZnO$, demonstrated robust stability under the continuous flow-enabled system, maintaining activity for over 100 h with minimal degradation. Specifically, $TiO_2$ exhibited sustained stability with a total reduction production rate of 825 μmol/g·h, a substantial improvement compared to the typical few hours of stability observed in flow-less batch systems (Fig. 2a). As shown in Fig. 2a and Supplementary Fig. 7, the production rate in the flow-less system exhibits a sharp decline within a few hours, followed by saturation. A decrease below 20% of the initial performance was regarded as a loss of catalytic activity, consequently, no further measurements were conducted. Similarly, $ZnO$ retained stability over extended periods (~100 h), highlighting the effectiveness of flow conditions in stabilizing UV-light-driven photocatalytic materials (Fig. 2b).

For visible-light-driven photocatalysts, $CdS$ and $C_3N_4$ (Fig. 2c, d), the flow-enabled system also provided significant stability improvements, but with some material-specific variations. $CdS$ displayed relatively faster degradation compared to $TiO_2$ and $ZnO$, retaining initial activity for approximately 60 h before noticeable deactivation occurred (Fig. 2c). This behavior might be attributed to the intrinsic properties of $CdS$, such as its susceptibility to photo-corrosion and surface oxidation[15,39], which are well-documented challenges in both flow-less batch and flow-enabled systems. Similarly, $C_3N_4$ demonstrates relatively limited stability due to its intrinsic instability to photo-corrosion under light exposure, leading to deactivation over time[17] (Fig. 2d). In contrast, under the flow-less conditions, all photocatalysts exhibited a rapid and severe loss of activity. Specifically, $CdS$ and $C_3N_4$ lost 50% of their initial activity within 2 h and dropped below 20% within 4 h, indicating significant loss of catalytic function. These degradation trends, observed in both flow-less and flow-enabled systems, emphasize the fundamental material-dependent limitations that cannot be entirely mitigated by flow control alone. Overall, these findings suggest that while continuous flow conditions substantially alleviate many stability issues by enhancing mass transport and preventing the accumulation of reaction intermediates, the intrinsic properties of the photocatalysts remain important in determining long-term operational performance.

To demonstrate the extended long-term stability achieved by our flow-enabled system, we conducted a 25-day continuous $CO_2$ reduction experiment using $TiO_2$, which exhibited the best stability and activity in our system under optimized flow conditions of 10 sccm $CO_2$ gas and 10 mL/min $H_2O$ (Fig. 2e). In this continuous flow-enabled photocatalytic system, each flow parameter significantly influences both the production rate and selectivity of the $CO_2$ reduction reaction[32,40]. In this work, we optimized the flow parameter to predominantly generate CO as the primary product (Supplementary Fig. 8). Under these optimized conditions, CO was the only detectable product; no liquid-phase products (e.g., $CH_3OH$, $HCOOH$) or other gas-phase byproducts (e.g., $CH_4$) were observed above the detection limit of our system. During the first two weeks, the system maintained its initial activity; however, by the 15th day, activity declined to approximately 80% of its initial level, followed by a further decrease thereafter. Notably, this level of photocatalytic stability represents a highly competitive continuous operational duration for $CO_2$ reduction using $TiO_2$ among reported systems[21]. From a practical continuous-

production perspective, over equivalent operation periods the accumulated CO output in the flow-enabled $TiO_2$ reactor substantially exceeds that of the flow-less batch system (Supplementary Fig. 9). Furthermore, performance is design-tunable and can be fine-tuned through rational cell-geometry design and flow rate optimization (Supplementary Fig. 10), providing additional avenues to enhance both activity and selectivity.

To establish quantitative redox coupling, we measured oxidative products after $CO_2$ purging ($O_2$ measured by GC−TCD; $H_2O_2$ measured by iodometric UV−vis in Supplementary Fig. 11). Due to photocatalytic water oxidation reactions (WOR), a significant amount of $H_2O_2$ can be produced along with $O_2$, and the yield strongly depends on the reaction environment and the properties of the photocatalyst[41,42]. As a result, hole equivalents ($O_2$, $H_2O_2$) matched electron equivalents in CO/$CH_4$ on $TiO_2$ ($e^-/h^+ \approx 1$) and accounted for the hole flux on g-$C_3N_4$, confirming that photogenerated holes are efficiently consumed in productive water oxidation rather than catalyst degradation, thereby preserving catalyst integrity (Supplementary Figs. 12, 13).

Based on our findings, we propose that the continuous flow of $CO_2$ and $H_2O$ in natural photosynthesis plays a critical role in maintaining long-term stability, emphasizing the importance of mimicking these natural transport processes in artificial photocatalytic systems[43]. In plants, water is transported from roots to leaves via transpiration, while $CO_2$ diffuses through stomata to reach chloroplasts; additionally, wind enhances $CO_2$ uptake by reducing boundary layer resistance[44]. These dynamic transport mechanisms ensure a consistent supply of reactants and efficient removal of products, contributing to the sustained efficiency and longevity of natural photosynthesis. Although the introduction of continuous flow in our artificial photocatalytic system significantly enhanced long-term stability, its durability still remains inferior to that of natural photosynthetic systems. Therefore, we suggest that a more systematic integration of flow regulation with catalyst design is essential to further improve the long-term operational stability of photocatalytic $CO_2$ reduction, ultimately aiming to emulate the considerable durability observed in nature.

## Surface mechanism study

The significant enhancement in stability observed in our continuous flow-enabled photocatalytic system can be attributed to the combined effects of $CO_2$ and $H_2O$ flow over the photocatalytic surface, which optimize the reaction environment and prevent catalyst deactivation. To elucidate the individual contribution of $CO_2$ and $H_2O$ flow to photocatalyst stability, we systematically varied flow conditions during a 100-h reaction period: (i) $CO_2$ flow without $H_2O$ flow, (ii) reduced $CO_2$ flow with $H_2O$ flow (iii) both $CO_2$ and $H_2O$ flows maintained (Fig. 3a, b).

As shown in Fig. 3a, maintaining both optimized $CO_2$ gas flow (10 sccm) and $H_2O$ flow (10 mL/min) resulted in the highest stability, effectively mitigating catalyst deactivation. In the absence of $H_2O$ flow, the system was initially filled with $H_2O$, but no active circulation was maintained. Under these conditions, the reaction initially displayed a gradual decline in activity, which then plateaued at a lower level. This suggests that while $CO_2$ flow alone can sustain limited activity, the absence of $H_2O$ flow compromises the formation of a stable triple-phase interface on the catalyst surface, leading to diminished initial activity. On the other hand, when $CO_2$ flow was significantly reduced (1 sccm) but not entirely eliminated, the system retained moderate initial activity. However, a progressive decline was observed over time. This behavior indicates that $CO_2$ gas flow plays a more prominent role in sustaining activity during extended reactions, particularly after the initial stabilization phase. It is likely that $H_2O$ flow has a greater influence on the initial reaction activity, possibly due to infiltration of $H_2O$ into the gas diffusion layer under no-flow condition, which adversely affect the formation and maintenance of the triple-phase interface[32,45,46]. As the reaction progresses and the system stabilizes, $CO_2$ gas flow becomes a critical factor in maintaining long-term

stability by ensuring a continuous supply of reactants to the photocatalyst surface while potentially facilitating the removal of intermediates and products, an effect that will be further examined in the following sections. These findings emphasize the importance of optimizing both $H_2O$ and gas flow rates to achieve long-term stability of photocatalytic $CO_2$ reduction.

To investigate the detailed impact of $CO_2$ and $H_2O$ flow on photocatalyst deactivation mechanism, we hypothesized that carbon species from side reactions accumulate on the photocatalyst surface during the reaction (Fig. 3b). This failure mechanism, involving carbon species accumulation, has been proposed previously[47], showing that flooding of GDEs is caused by polymerization of minor products on their surfaces. Several studies have reported that reaction intermediates and carbon byproducts are adsorbed on the catalyst surface, inhibiting photocatalytic activity[35,48,49]. In our case, we believe that the presence or absence of $CO_2$ and $H_2O$ flow influences the deposition of carbon species, leading to differences in stability.

First, various analyses were conducted to identify stability degradation factors. A gradual decline in activity after the initial stage can result from chemical or structural change in the catalyst during the photocatalytic reaction. X-ray photoelectron spectroscopy (XPS) analysis in Supplementary Fig. 14 of the photocatalyst surface after 25 days (Fig. 2e) revealed that the main Ti $2p$ and O $1s$ peaks remained unchanged, indicating that structural changes in the catalyst itself can be ruled out. As will be discussed in more detail later, the increase in the C–C peak in the C $1s$ spectrum further supports surface modifications with carbon species. Scanning electron microscopy (SEM) and high-resolution transmission electron microscopy (HRTEM) of the catalyst layer before and after the reaction (Supplementary Figs. 15 and 16) revealed no significant changes in the loading or structure of the catalyst, indicating that the decline in activity is not caused by physical detachment of the catalyst but rather by surface modification phenomena or other factors affecting the catalyst's reactivity.

Therefore, we shifted our focus from the catalyst itself to the microenvironment and the catalyst surface. A strong relation between stability and gas–liquid flow can be attributed to carbon species formation derived from intermediates poisoning, which is effectively addressed through desorption mechanisms. By preventing the re-adsorption of deactivating species on the photocatalyst surface, the system maintains high catalytic activity over extended operational periods. Experimental results validate the flow-enabled system as a robust and enhanced platform for photocatalytic $CO_2$ reduction, demonstrating notable advancements in both performance and longevity compared to traditional system without flow media (Fig. 3b). In the reaction environment without adequate $CO_2$ and $H_2O$ flow, mass transfer occurs primarily through diffusion. As the photocatalytic reaction progresses, products and intermediates accumulate on the catalyst surface. Even if desorption occurs, localized products, such as CO in the case of $TiO_2$, exhibit higher partial pressure near the catalyst surface compared to the bulk environment. This distribution phenomenon, due to the simultaneous occurrence of oxidation and reduction reactions in photocatalysis, can lead to undesired reactions[33,34], ultimately resulting in lower production efficiency. Additionally, the accumulation of high concentrations of products near the catalyst accelerates the attainment of reaction equilibrium, reducing reaction rates and increasing the probability of further undesired reactions. Products that are not timely desorbed may adhere to the surface, causing poisoning effects that reduce the effective catalytic surface area, thereby diminishing initial activity and contributing to gradual deactivation. In contrast, the flow environment depicted in the 3rd image of Fig. 3b creates a microenvironment where external flow continuously supplies reactants and removes products. In this configuration, products generated near the catalyst surface are swiftly swept away by the flux, preventing their accumulation and

allowing them to mix into the bulk environment. This dynamic prevents undesired reactions and effectively inhibits product fixation on the catalyst surface. As a result, the flow environment not only enhances reaction efficiency but also mitigates poisoning effects, maintaining the catalyst's activity over extended operational periods.

To verify our hypothesis, we conducted detailed surface analyses of $TiO_2$ photocatalysts subjected to different reactor designs (Fig. 3c-g). After 100 h of reaction, the reactors were disassembled, and the color of the photocatalyst layers was examined. Compared to the unreacted reference catalyst (Supplementary Fig. 17a), the catalyst from the flow-enabled cell system exhibited minimal color change (Supplementary Fig. 17b), whereas significant discoloration was observed in the flow-less batch system catalyst (Supplementary Fig. 17c). Additionally, after a 25-day long-term reaction (Fig. 2e), the catalyst from the flow-enabled system showed noticeable but less pronounced color changes (Supplementary Fig. 17d). Given that no structural change were detected in the $TiO_2$ itself (Supplementary Fig. 14), these color changes suggest that surface modifications, such as the adsorption of intermediate, are correlated with the diminishing catalytic activity[50,51].

XPS analysis of the C $1s$ bonds was conducted to further investigate these surface changes (Fig. 3c-e). The reference photocatalyst layer (Fig. 3c) exhibited dominant C-F bonds from the PTFE GDL substrate, along with minor C-C, C=O, and O-C-O bonds[52] (the C-C/C-$F_2$ ratio of the reference and bare PTFE is ~0.73 in Supplementary Fig. 18). In the flow-enabled sample (Fig. 3d), the C-$F_2$ and C-C peaks were similar to those of the reference (the C-C/C-$F_2$ ratio ~0.72), indicating minimal surface modifications. However, the flow-less system sample (Fig. 3e) showed a significant increase in the C-C bond intensity, surpassing the C-$F_2$ bond (the C-C/C-$F_2$ ratio ~1.71). This result suggests that carbon-carbon bonds dominated the surface, likely masking the C-$F_2$ signal from the PTFE substrate due to carbonaceous species accumulating on the catalyst surface.

To further examine the nature of the carbon species, Fourier transform infrared (FTIR) analysis was performed (Fig. 3f). The PTFE substrate's characteristic peaks were observed consistently at ~500 $cm^{-1}$ and 1000–1300 $cm^{-1}$ in all samples, with anatase $TiO_2$ absorption peaks present in the 400–800 $cm^{-1}$ range[53]. However, in the 1400–1800 $cm^{-1}$ region (inset of Fig. 3f), distinct differences emerged. The reference and flow-enabled system samples showed similar absorption patterns, while the flow-less system sample displayed a significant increase in peaks corresponding to C=C (~1450 $cm^{-1}$), benzene-related C-C (~1546 $cm^{-1}$), and C=C (~1639 $cm^{-1}$) bonds. Additionally, peaks for C=O bonds were notably enhanced. These results confirm the formation of carbonaceous complexes, likely resulting from the accumulation of reaction intermediates or products on the photocatalyst surface in the flow-less system.

To identify the specific carbonaceous species, matrix-assisted laser desorption ionization-time-of-flight mass spectrometry (MALDI-ToF MS) analysis was performed (Fig. 3g). Previous studies have reported the detection of butadiene at 57 m/z on gas diffusion electrodes for $CO_2$ electroreduction[47]. Similarly, in our system, a peak at 57 m/z was observed exclusively in the flow-less system sample, while it was absent in both the reference and flow-enabled system samples. This finding indicates the accumulation of $C_4$ or larger carbon species on the catalyst surface in the flow-less system, further supporting the hypothesis that carbon accumulation is a key factor in catalytic deactivation under batch conditions. Accumulation of carbonaceous species was also assessed by evaluating changes in surface hydrophilicity after prolonged reaction conditions, using contact angle measurements (Supplementary Fig. 19). The reference catalyst exhibited a contact angle of 133°, indicative of the hydrophobic nature of the PTFE substrate. The flow-enabled system sample showed a slightly reduced contact angle of 127°, suggesting minimal surface changes. However, the flow-less system sample displayed a significantly reduced contact

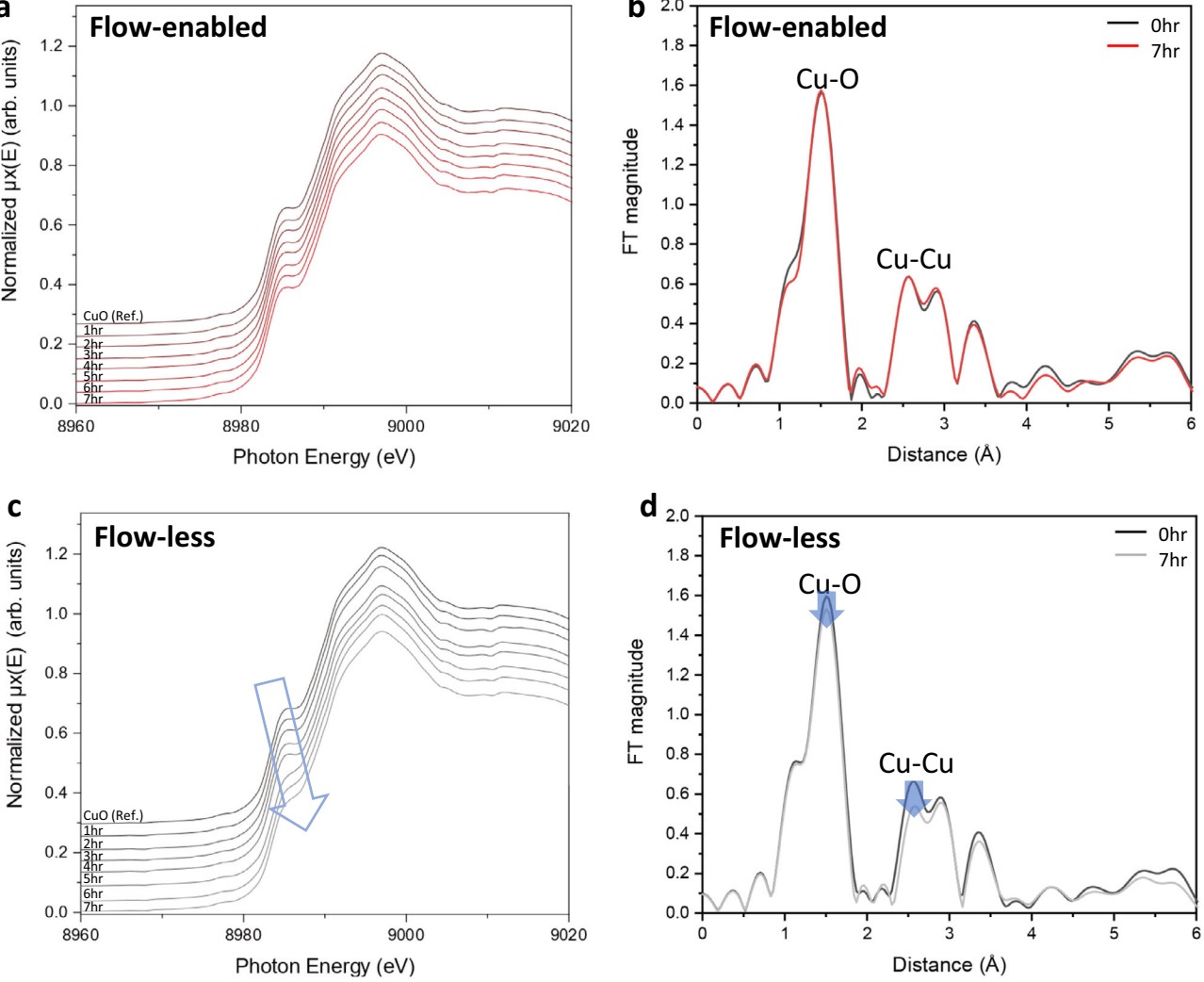

**Fig. 4 | In-situ XAFS analysis of CuO photocatalyst under flow-enabled and flow-less conditions. a** In-situ Cu K-edge XANES spectra of CuO during a 7-h reaction under continuous flow conditions. **b** Fourier transform of the Cu K-edge EXAFS spectra under flow conditions, revealing no observable structural changes before and after the reaction. **c** In-situ Cu K-edge XANES spectra of CuO during a 7-h reaction under gas-phase flow-less conditions, showing progressive broadening of the Cu edge indicative of structural degradation (arrow). **d** Fourier transform of the Cu K-edge EXAFS under flow-less conditions, showing reduction in the Cu-O and Cu-Cu coordination peaks (arrows), consistent with significant changes in the bonding structure. All measurements were performed under 100 mW cm⁻² illumination from an Xe lamp connected with an optical fiber.

angle of 106°, implying increased surface hydrophilicity. This is consistent with the accumulation of carbonaceous species or surface modifications resulting from extended reactions in the flow-less system environment.

These comprehensive surface analyses reveal that main reason for stability issue of flow-less system is from significant carbon accumulation and surface modifications, leading to catalytic deactivation. In contrast, the system with both $CO_2$ and $H_2O$ flows effectively mitigates such issues, maintaining a stable reaction environment and preserving the photocatalyst's surface properties. These findings highlight the detrimental effects of the flow-less system conditions and the critical role of flow dynamics in sustaining photocatalytic activity over extended periods.

The deactivating effect of carbon accumulation on the photocatalyst surface was confirmed, and an experiment was conducted to determine whether activity could be restored by removing the accumulated carbon. To facilitate carbon buildup under conditions of inefficient mass transfer, the photocatalyst was coated with a thick layer in the flow-enabled system (catalyst loading rate >4 mg/cm²), leading to a more rapid decline in activity. As shown in Supplementary

Fig. 20, when carbon accumulated on the surface, a drop of dilute sulfuric acid solution (0.05 M $H_2SO_4$) applied to the catalyst on a 100 °C hot plate effectively oxidized and removed the surface carbon within 1 min. Furthermore, after this cleaning treatment, the catalyst was reintroduced into the reactor, and its initial activity was successfully restored (Supplementary Fig. 21). These results indicate that the deactivating carbon species were not permanently immobilized but rather consisted of acid-soluble carbon deposits. Moreover, even if the photocatalyst undergoes deactivation in the flow-enabled system, it can be easily regenerated through a simple cleaning process, demonstrating the practical advantage of reusability in long-term applications.

### Atomic-level study

Our mechanism for long-term photocatalytic stability with $CO_2$ and $H_2O$ flow was further investigated by observing chemical state analysis at the atomic level. We performed synchrotron-based X-ray absorption near-edge structure (XANES) and Extended X-ray Absorption Fine Structure (EXAFS) analysis. XANES is particularly suitable for probing local electronic and coordination environments, allowing us to assess

whether the bulk structure of the photocatalyst remains intact under different reaction conditions. Ex-situ Ti K-edge XANES and EXAFS analyses were performed on pristine $TiO_2$ samples after 100 h of photocatalytic $CO_2$ reduction under both flow-enabled and flow-less conditions. The Ti K-edge XANES spectra (Supplementary Fig. 22a) showed no noticeable shifts in edge position or white-line intensity between the two systems, indicating that the oxidation state and electronic structure of Ti remained stable during long-term operation. This observation is consistent with XPS measurements (Supplementary Fig. 14a, b), which also revealed negligible variation in the Ti $2p$ binding energies between the pristine and reacted samples, further confirming the chemical state stability of Ti during long-term operation.

In contrast, the Fourier-transformed EXAFS spectra (Supplementary Fig. 22b) revealed subtle but noticeable differences in local atomic arrangements, particularly under flow-less conditions. Specifically, for flow-less sample, the Ti-O bond length exhibited a slight increase, whereas the Ti-Ti bond distance became marginally shorter compared to both the reference and flow-enabled samples. This combination of bond length expansion and contraction suggests the presence of local lattice distortion, possibly induced by surface stress or coordination rearrangements[54–56]. These distortions are likely caused by the accumulation of reaction intermediates or carbonaceous species in the absence of continuous flow, which can alter the local structural environment without triggering bulk phase transitions. Supporting this interpretation, surface-sensitive techniques such as C *1s* XPS and FTIR analyses also indicated pronounced surface modifications under the flow-less conditions. On the other hand, $TiO_2$ samples from the flow-enabled system retained interatomic distances nearly identical to those of the unreacted reference, highlighting the protective role of dynamic gas and liquid flow in maintaining both the chemical and structural integrity of the catalyst. These findings underscore that while the bulk structure of $TiO_2$ remains stable, local distortions at the atomic level can arise in stagnant systems and contribute significantly to long-term catalyst deactivation.

While ex-situ XAFS analyses of $TiO_2$ revealed discernible perturbations in local coordination environments under the flow-less conditions, the extent of structural reorganization was relatively minor, and no significant phase transitions or changes in oxidation state were detected. This limited degree of atomic-level restructuring, likely attributable to the intrinsic crystallographic stability of $TiO_2$, presented challenges in fully resolving the dynamic structural responses to different flow environments[57]. To overcome this limitation and gain deeper insight into the role of flow in modulating catalyst stability, we extended our investigation to CuO, which is a well-known material to undergo more pronounced local structural transformations under photocatalytic conditions[58–60]. CuO exhibits coordination changes and redox behavior under reaction conditions, which are effectively tracked by XAFS analysis[61–63]. These characteristics make CuO a suitable model system for evaluating how flow conditions influence not only catalytic performance but also the structural integrity and active state of the photocatalyst in real time (Fig. 4). Both CuO samples were prepared identically, with minor variations attributed to differences in electrode preparation, as confirmed by identical pre-reaction Cu K-edge XANES spectra (Supplementary Fig. 23).

To enable these real-time measurements, we successfully configured an in-situ setup with a dedicated flow photocatalytic system compatible with synchrotron X-ray analysis (Supplementary Fig 24). The custom-designed reactor was engineered to allow simultaneous light irradiation, gas flow, and X-ray measurements while maintaining a controlled reaction environment. A Xe lamp was used as the light source, and light was delivered directly to the photocatalyst layer via an optical fiber. The reaction surface was continuously irradiated with a light intensity of 100 mW/cm², covering X-ray exposure area. The gas flow system was integrated with a mass flow controller to ensure

precise delivery of $CO_2$ (10 sccm) and $H_2O$ flow (10 mL/min) to the photocatalyst surface. To facilitate X-ray measurements, the reactor included a dedicated X-ray window located at the gas flow component. X-rays were directed at an angle perpendicular to the gas flow layer, penetrating the PTFE substrate to reach the photocatalyst layer coated on the inner surface of the membrane. The fluorescence emissions from the photocatalyst surface were captured by a silicon detector positioned at an appropriate angle, enabling real-time tracking of Cu bonding changes and structural evolution. For the flow-less reaction, the reactor was purged with $CO_2$ containing $H_2O$ vapor at a flow rate of 100 sccm for 1 h, after which all connected lines were closed, and the system was exposed to Xe light irradiation at an intensity of 100 mW/cm².

Under flow-enabled conditions, the Cu K-edge XANES spectra (Fig. 4a) remained virtually unchanged over 7-h reactions, with no significant shifts in edge position or white-line intensity. This indicates that the oxidation state and coordination geometry of Cu were well preserved[64,65]. Consistent with this, the EXAFS spectra (Fig. 4b) also showed stable Cu–O and Cu–Cu coordination peaks with consistent amplitudes and radial distances, confirming the structural robustness of CuO in the presence of continuous reactant flow. Under flow-less conditions, however, the Cu K-edge XANES spectra (Fig. 4c) exhibited progressive spectral changes over time. In particular, a gradual decrease in the pre-edge intensity was observed, which may reflect increasing structural disorder or a loss of well-defined coordination symmetry around Cu centers[66]. The corresponding EXAFS data (Fig. 4d) also revealed a notable reduction in the amplitude of both Cu–O and Cu–Cu coordination shells. The EXAFS spectra showed clear decreases in both coordination peaks, indicating a loss of local structural ordering around the Cu centers. This structural degradation under flow-less conditions is likely driven by the accumulation of reaction intermediates, such as carbohydroxides and other surface-bound residues, which alter the local bonding environment around Cu[60].

The structural trends observed by XAFS were strongly correlated with catalytic performance, as shown in the Supplementary Fig 25. Under flow-enabled conditions, CuO maintained steady photocatalytic $CO_2$ reduction activity over the course of 30 h, with minimal loss in product formation rate. In contrast, the flow-less system exhibited rapid performance decay, with a significant drop below 20% in activity within the first several hours. This performance degradation is consistent with the progressive loss of local coordination order revealed by in-situ XAFS under flow-less conditions, suggesting that the accumulation of reaction byproducts not only distorts the structural environment but also suppresses active site functionality. These findings highlight the critical role of continuous flow in mitigating surface poisoning and preserving both the structural and functional stability of CuO-based photocatalysts.

Taken together, the results from both $TiO_2$ and CuO systems reveal that flow-enable system is a broadly applicable strategy for preserving photocatalyst integrity under prolonged operation. While $TiO_2$ exhibited only subtle coordination-level perturbations due to its intrinsically rigid lattice, CuO responded with more pronounced structural changes that were sensitively captured by in-situ XAFS. Despite the differences in structural reactivity, both systems clearly benefited from continuous flow, which suppressed the accumulation of surface-bound intermediates and mitigated local coordination disorder. These findings suggest that beyond enhancing mass transport, flow conditions play a vital role in maintaining the chemical and structural stability of heterogeneous catalysts during operation.

## Discussion

This study demonstrates the potential of a continuous flow photocatalytic system in overcoming stability challenges for photocatalytic $CO_2$ reduction. The flow-enabled system effectively mitigated product

accumulation and undesired reactions, ensuring sustained activity for over 100 h with basic photocatalysts and up to 15 days for CO production. Comprehensive surface and structural analyses reveal that catalyst deactivation in the system without flows is primarily driven by the accumulation of carbon species, whereas the continuous flow-enabled approach maintains minimal surface modifications, preserving catalytic integrity. In-situ XAFS analysis further confirms the enhanced stability of CuO under flow conditions, exhibiting negligible changes in bonding environments compared to significant degradation in flow-less system. In conclusion, the continuous flow system provides a robust platform for scalable and sustainable photocatalysis, addressing long-term stability challenges. These findings establish a foundation for future innovations in photocatalytic reactor design and operation. We believe that a more systematic flow design of $CO_2$ and $H_2O$, combined with appropriate photocatalysts, will enable long-term stable photocatalytic $CO_2$ reduction, comparable to the considerable resilience of natural photosynthesis systems.

# Methods
## Materials
Methanol (MeOH, 99.9%), a Nafion 117 solution (~5 wt%), zinc oxide (ZnO, <50 nm particle size), copper oxide (CuO, <50 nm particle size) and cadmium sulfide (CdS, 99.995%) were purchased from Sigma-Aldrich. $TiO_2$ powder (P25, Degussa) and $C_3N_4$ powder were used as received. The gas diffusion layer (GDL; Porex PM21M) was purchased from Fuel Cell Store. All chemicals were used as received without further purification.

## Preparation of catalyst layers
To prepare the catalyst ink used for every photocatalytic sample, 10 mg of catalyst powder was sonicated for 30 min in 10 mL of methanol containing 20 μL of Nafion solution. A 500 μL aliquot of the resulting ink was then spray-coated onto the gas-diffusion layer (GDL) and the coated film was left to dry overnight.

## Photocatalytic reaction in the flow-less system
For the flow-less photocatalytic $CO_2$-reduction experiment, a 50 mL stainless-steel reactor (SUS) equipped with a top quartz window was used. Illumination came from a 300 W xenon lamp, delivering 300 mW cm$^{-2}$ of light inside the chamber. A 1 cm$^2$ catalyst specimen was positioned on the reactor floor, after which 4 mL of $H_2O$ was added. The vessel was then thoroughly purged with $CO_2$ to saturate both the catalyst surface and the headspace with $CO_2$ and water vapor before irradiation. Photocatalytic reactions were allowed to occur for each set time, and the amounts of the products obtained were analyzed using a gas chromatograph (GC; Agilent 7890GC) connected to the reactor.

## Photocatalytic reaction in the flow-enabled system
The flow-enabled system[32] was constructed entirely from chemically inert stainless steel (SUS), comprising a quartz window housing, a $H_2O$ flow plate, and a gas flow plate, each containing a 1 cm diameter circular hole at the center. The photocatalyst-coated gas diffusion layer (GDL) was positioned between the two flow plates with the photocatalyst coated side facing the quartz window, so that light from the 300 W Xe lamp passed through the water layer before reaching the catalyst surface. The light intensity was measured using an Optical Power Meter (Newport) and adjusted to either 300 mW cm$^{-2}$ or 100 mW cm$^{-2}$ depending on the experimental conditions. Gas pressure was controlled using a low-pressure regulator installed upstream of the gas flow plate inlet to set the desired pressure, while a pressure gauge placed downstream provided real-time monitoring of the in-system pressure. A needle valve positioned after the pressure gauge was used to adjust the gas outflow rate, which was continuously confirmed by a downstream mass flow meter (MFM). Water flow was controlled by a peristaltic pump configured as a closed loop without an external reservoir, which balanced the liquid-side pressure against the gas-side pressure. Prior to each experiment, the reactor was assembled and leak-tested. Start-up procedure involved first establishing $H_2O$ circulation at 10 mL min$^{-1}$ to wet the catalyst surface, followed by gradual introduction of $CO_2$ flow from 1 to 10 sccm over 5 min while monitoring the system pressure (maintained at 1.2 bar). This gradual ramp-up prevents sudden pressure changes that could disrupt the gas-liquid interface or cause flooding. During operation, the balance between gas and liquid phases was monitored by observing bubbles and flooding. Gaseous products were continuously sampled and quantified by an in-line gas chromatograph (GC; Agilent 7890GC) at each designated time interval. Prior to each run, pure $CO_2$ was introduced into the system to establish a baseline and confirm the absence of residual gases.

## Product analysis
The production rate in the flow-less system was determined according to the following equations:
For the CO production rate:

$$yCO = \frac{\text{CO produced(mol)}}{\text{catalyst amount(g)} \cdot \text{reation time(hr)}}$$
$$= \frac{\text{fraction of CO(\%)} \cdot \text{Reactor Pressure(atm)} \cdot \text{Reactor volume(L)}}{0.082\left(\frac{\text{atm} \cdot \text{L}}{\text{mol} \cdot \text{K}}\right) \cdot \text{reactor Tempaerature(K)} \cdot \text{catalyst amount(g)} \cdot \text{reation time(hr)}}$$

In the case of the flow-enabled system, the production rate was calculated by considering the total number of moles of gas entering the GC system and the ratio of the products formed therein.
For the CO production rate:

$$yCO = \frac{\text{CO produced(mol)}}{\text{catalyst amount(g)} \cdot \text{reation time(hr)}}$$
$$= \frac{\text{fraction of CO(\%)} \cdot n_{total}}{\text{catalyst amount(g)} \cdot \text{reation time(hr)}} \text{ and}$$
$$n_{total} = \frac{\text{atmospheric pressure(atm)} \cdot \text{injected sample volume(L)}}{0.082\left(\frac{\text{atm} \cdot \text{L}}{\text{mol} \cdot \text{K}}\right) \cdot \text{room temperature(K)}}$$
$$= \frac{\text{atmospheric pressure} \cdot \text{gas flow rate}\left(\frac{\text{ml}}{\text{min}}\right) \cdot \text{gas flow time(min)}}{0.082\left(\frac{\text{atm} \cdot \text{L}}{\text{mol} \cdot \text{K}}\right) \cdot \text{room temperature(K)}}$$

By combining the above two formulas, as the reaction time was the same as the time of the gas flow in a continuous system:

$$y_{co} = \frac{\text{fraction of CO(\%)} \cdot \text{atmospheric pressure} \cdot \text{gas flow rate}\left(\frac{\text{ml}}{\text{min}}\right)}{0.082\left(\frac{\text{atm} \cdot \text{L}}{\text{mol} \cdot \text{K}}\right) \cdot \text{room temperature(K)} \cdot \text{catalyst amount(g)}}$$

## Quantitative analysis of $CO_2$ mass transfer enhancement
To quantitatively validate the mass-transfer enhancement in our flow-enabled photocatalytic system, we conducted $CO_2$ uptake experiments under strong-sink conditions. This approach provides direct evidence of improved $CO_2$ transport compared to conventional flow-less batch systems. CO2 uptake rates were measured using 0.10 M NaOH solution as a strong $CO_2$ sink, with phenolphthalein as the endpoint indicator. Both flow-enabled and flow-less reactors were tested under identical conditions: 1) Gas-liquid contact area: 0.785 cm$^2$ (1 cm diameter opening), 2) Temperature: 25 °C, 3) $CO_2$ pressure: 1.2 bar, and 4) No photocatalyst or light irradiation.

For the flow-enabled system, 10 mL of NaOH solution was circulated while $CO_2$ gas flowed at 10 sccm. For the flow-less system, 1 mL of NaOH solution was placed in a sealed chamber with $CO_2$ headspace. The endpoint was determined when the phenolphthalein indicator turned colorless (pH ≈ 8.2).

The $CO_2$ uptake was calculated based on the neutralization reaction:

$CO_2 + OH^- \rightleftarrows HCO_3^-$ (at phenolphthalein endpoint)

For a Flow-enabled system:

Time to endpoint: 2 h

NaOH volume: 10 mL

Moles of $CO_2$ absorbed: $n_{CO_2} \approx C_{NaOH} \times V_{NaOH} = 0.10\,M \times 0.010\,L = 1.0 \times 10^{-3}\,mol$

For Flow-less system:

Time to endpoint: 3 h

NaOH volume: 1 mL

Moles of $CO_2$ absorbed: $n_{CO_2} = 0.10\,M \times 0.001\,L = 1.0 \times 10^{-4}\,mol$

The area- and time-normalized $CO_2$ flux (J) was calculated as:

$$J = nCO2/(A \times t)$$

Where A is the gas–liquid contact area and t is the time to endpoint.

Flow-enabled: $J = (1.0 \times 10^{-3}\,mol)/(7.85 \times 10^{-5}\,m^2 \times 7200\,s) = 1.76 \times 10^{-3}\,mol\,m^{-2}\,s^{-1}$

Flow-less: $J = (1.0 \times 10^{-4}\,mol)/(7.85 \times 10^{-5}\,m^2 \times 10{,}800\,s) = 1.18 \times 10^{-4}\,mol\,m^{-2}\,s^{-1}$

The apparent permeance (P) was calculated as:

$$P = J/(y_{co2} \times p)$$

Where $y_{co2} = 1$ (pure $CO_2$) and p = 1.2 bar for both systems.

Flow-enabled: $P = 1.47 \times 10^{-8}\,mol\,m^{-2}\,s^{-1}\,Pa^{-1}$

Flow-less: $P = 9.8 \times 10^{-10}\,mol\,m^{-2}\,s^{-1}\,Pa^{-1}$

Enhancement factor: $P_{flow\text{-}enabled}/P_{flow\text{-}less} = J_{flow\text{-}enabled}/J_{flow\text{-}less} \approx 15$

The quantitative $CO_2$ uptake experiments demonstrate that our *flow-enabled* photocatalytic system achieves a 15-fold enhancement in $CO_2$ mass transfer compared to conventional *flow-less* batch systems. This dramatic improvement in mass transport is a key factor contributing to the superior long-term stability and sustained activity observed in our *flow-enabled* photocatalytic $CO_2$ reduction system.

### Characterization

Surface morphology was characterized by scanning electron microscopy (SEM, Hitachi SU8230). For ex-situ analyses, the photocatalyst layer was removed from both the flow-enabled and flow-less reactors after testing, oven-dried at 60 °C in the dark, and then evaluated for surface changes using X-ray photoelectron spectroscopy (XPS), Fourier-transform infrared spectroscopy (FTIR), static contact-angle measurements, and matrix-assisted laser desorption/ionization time-of-flight mass spectrometry (MALDI-ToF).

### Synchrotron-based X-ray absorption spectroscopy measurements

X-ray absorption near-edge structure (XANES) and Extended X-ray Absorption Fine Structure (EXAFS) were acquired at the 10C nanoprobe XAFS beamline (BL10C) of Pohang Light Source II. Prior to each run, the energy scale was calibrated using a metal-foil reference. Ex-situ spectra were acquired in both transmission and fluorescence modes and processed with the Demeter software package.

For in-situ flow-enabled studies, we employed a custom reactor that enabled simultaneous light irradiation, gas delivery, and X-ray probing under well-controlled conditions. A xenon lamp supplied illumination, which was guided to the catalyst layer through an optical fiber and maintained at 100 mW cm$^{-2}$. The gas flow system was integrated with a mass flow controller to ensure precise delivery of $CO_2$ (10 sccm) and $H_2O$ flow (10 mL/min) to the photocatalyst surface. To facilitate X-ray measurements, the reactor included a dedicated X-ray window located at the gas flow component. X-rays were directed at an angle perpendicular to the gas flow layer, penetrating the PTFE substrate to reach the photocatalyst layer coated on the inner surface of the membrane. The fluorescence emissions from the photocatalyst surface were captured by a silicon detector positioned at an appropriate angle, enabling real-time tracking of Cu bonding changes and structural evolution.

For the in-situ flow-less reaction, the reactor was purged with $CO_2$ containing $H_2O$ vapor at a flow rate of 100 sccm for 1 h, after which all connected lines were closed, and the system was exposed to Xe light irradiation at an intensity of 100 mW/cm$^2$.

For in-situ measurement, the custom-designed reactor was engineered to allow simultaneous light irradiation, gas flow, and X-ray measurements while maintaining a controlled reaction environment. A Xe lamp was used as the light source, and light was delivered directly to the photocatalyst layer via an optical fiber. The reaction surface was continuously irradiated with a light intensity of 100 mW/cm$^2$, covering X-ray exposure area. The gas flow system was integrated with a mass flow controller to ensure precise delivery of $CO_2$ (10 sccm) and $H_2O$ flow (10 mL/min) to the photocatalyst surface. To facilitate X-ray measurements, the reactor included a dedicated X-ray window located at the gas flow component. X-rays were directed at an angle perpendicular to the gas flow layer, penetrating the PTFE substrate to reach the photocatalyst layer coated on the inner surface of the membrane. The fluorescence emissions from the photocatalyst surface were captured by a silicon detector positioned at an appropriate angle, enabling real-time tracking of Cu bonding changes and structural evolution. For the in-situ flow-less reaction, the reactor was purged with $CO_2$ containing $H_2O$ vapor at a flow rate of 100 sccm for 1 h, after which all connected lines were closed, and the system was exposed to Xe light irradiation at an intensity of 100 mW/cm$^2$.

## Data availability

All data generated in this study are provided in the Supplementary Information/Source Data file Source data are provided with this paper.

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

## Acknowledgements

This study was funded by the Saudi Aramco-KAIST $CO_2$ Management Center and the KAIST-UC Berkeley-VNU Climate Change Research Center grant (2021K1A4A8A01079356). Also, this work was supported by the National Research Council of Science & Technology (NST) grant by the Korea government (MSIT) (No. CAP25011-000) and KIST institutional project. XAFS experiments were conducted at 10 °C at Pohang Accelerator Laboratory of Pohang University of Science and Technology.

## Author contributions

H.J., C.K. and H.-T.J. conceived the idea and designed the experiments. H.J. prepared and optimized the *flow-enabled* system and performed the stability tests and surface analyses. H.J. and H.S.J. designed the *flow-enabled* and *flow-less* system for in-situ EXAFS measurements, and H.J., H.S.J., and M.G.K. analyzed the resulting data. A.J., I.G., and H.-T.J. secured funding for the work. C.K. and H.-T.J. supervised all steps of the research.

## Competing interests

The authors declare no competing interests.
