## [Transparent Peer Review file · Nature Communications]

Significant Stability Enhancement in Photocatalytic CO₂ Reduction via Flow- Driven Strategies

Corresponding Author: Professor Hee-Tae Jung

Version 0:

Reviewer comments:

Reviewer #1

(Remarks to the Author)

This work reports a previously underexplored yet promising strategy to overcome the major challenge of long-term stability in photocatalytic CO₂ reduction by introducing continuous CO₂ and H₂O flow during the process. Surface and structural analyses revealed that flow-less systems suffer from products/intermediates accumulation and undesired reactions on the photocatalyst surface, while flow-enabled systems maintain pristine catalytic surfaces. However, the authors did not compare their system with other flow systems. Furthermore, quantitative analysis of critical aspects, such as mass transfer efficiency, is lacking. Consequently, the manuscript still has significant limitations, and thus, I cannot recommend its publication in Nature Communications:

1. Please clarify the key advantages of the flow-enabled photocatalytic system developed herein over the existing flow systems (e.g., DOI: <https://doi.org/10.1002/anie.202302979>), specifically comparing mass transfer efficiency, resistance to carbon deposition, and reactive efficiency. Compared to your group's previous research on the "Continuous-flow reactor with superior production rate and stability for CO₂ reduction using semiconductor photocatalysts", which also utilized a flow system, the innovation of this paper is open to debate.
2. The paper uses CO production as the primary example, but the CO₂ reduction to CO reaction process is relatively simple. Is this flow strategy applicable to the production of complex hydrocarbon products with longer residence times and more complex reaction processes, such as ethanol and other multi-carbon products?
3. While the author mentions that "Introducing a continuous flow of CO₂ and H₂O across the photocatalyst surface addresses these limitations by enhancing mass transport", quantitative evidence for this mass transfer enhancement is lacking.
4. Would the flow control strategy affect the CO₂ conversion while improving stability?
5. The authors attribute the outstanding catalytic activity in part to the formation of a stable gas-liquid-solid (three-phase) interface. In this work, how does a flow-enabled photocatalytic system dynamically regulate the interfacial balance among the gas, liquid, and solid phases? Please elaborate on the synergistic interaction mechanism between CO₂ and H₂O at the three-phase interface.
6. Good stability is one of the basic conditions for industrial application. Does the flow strategy result in economically viable energy consumption and product separation costs in long-term operation, or does it sacrifice one for the other? It is recommended to supplement the discussion with a technical-economic analysis.
7. The source of CO production activity needs verification.
8. Although XPS/FTIR/MALDI-TOF have demonstrated that carbon species accumulation is the primary cause of deactivation, there is a lack of quantitative models or theoretical calculations to support the microscopic mechanisms of how flow dynamically inhibits carbon deposition, such as interfacial mass transfer kinetics and adsorption-desorption equilibrium. If DFT calculations or microkinetic simulations could be supplemented, this would significantly enhance the persuasiveness of the mechanism.
9. Are the catalysts used in Figures 4c and 4e the same? If so, why do the spectra at 0 h (before the reaction) show discrepancies in the relative intensities of certain peaks?
10. The study proposes that the flow strategy enhances catalyst stability by removing carbonaceous deposits. However, XPS results show the presence of C–C bonds on the catalyst surface even after prolonged reaction. Could the authors clarify how the flow system selectively removes specific types of carbon species?
11. The performance enhancement in the flow-enabled photocatalytic system is attributed to continuous reactant supply and efficient product removal. However, the mass transfer efficiency has not been quantitatively analyzed in the manuscript.
12. The authors should review and improve the writing conventions, such as the inconsistent formatting of number subscripts in the abstract.

Reviewer #2

(Remarks to the Author)

The work demonstrates the transformative potential of a continuous flow photocatalytic system in overcoming the stability challenges for photocatalytic CO₂ reduction. The developed flow-enabled system effectively mitigated product accumulation and undesired reactions, ensuring sustained activity for over one hundred hours with photocatalysts for CO production. Based on the surface and structural analyses, the continuous flow-enabled approach introduces minimal surface modifications, preserving catalytic integrity. The findings are interesting, and I think the manuscript can be published with the following several issues addressed.

1 Line 56: This approach mitigates mass transport limitations, prevents the accumulation of products and intermediates, and inhibits the formation of deactivating surface species.

During the reaction process, what effects do the reactants CO₂ and H₂O have on the catalyst?

How could the authors alleviate the effects of photo-corrosion under light exposure on catalysts?

2 Line 103: The product formation rate is up to 825 $\mu\text{mol}\cdot\text{g}^{-1}\cdot\text{h}^{-1}$ far exceeding the performance observed under batch conditions without CO₂ and H₂O flow. This represents a more than two order of magnitude enhancement in stability compared to previously reported TiO₂-based photocatalysts

In the flow-enabled platform, what is the pressure exerted by CO₂ and H₂O? Is the superior performance due to more CO₂ with higher local concentrations being involved in the reaction?

3 In this work, the authors optimized the flow parameter to predominantly generate CO as the primary product.

What are the respective single-pass CO₂ conversion rates in the flow-less batch and flow-enabled platform?

4 To enhance readability, please include schematic diagrams and actual photographs of the flow-enabled platform.

5 What is the temperature of the reaction system? How can the condensation of incoming H₂O vapor into water be prevented?

Line 221-223: It is likely that H₂O flow has a greater influence on the initial reaction activity.

If it only affects the initial reaction, why maintain a continuous flow of H₂O at 10 mL/min into the reaction system?

6 To further validate the catalyst stability after the reaction, please offer the TEM and HRTEM images of the (used?) catalyst.

7 For in-situ X-ray absorption spectroscopy, the CuO is used as the model catalyst. So what are the products when CuO is used as the photocatalyst in the flow-enabled platform? Is it CO? Please provide supplementary information.

Reviewer #3

(Remarks to the Author)

In the present study, the authors introduced the continuous CO₂ and H₂O flow during the photocatalytic CO₂ reduction process and achieved up to 50-fold improvement on the efficiency. The idea of using continuous flow is interesting. However, the presented results completely discarded the oxidation reactions accompanied with CO₂ reduction. The accompanied oxidation reaction, theoretically as water oxidation towards O₂ evolution, should be very difficult for the photocatalysts CdS, C₃N₄ and CuO. In most cases, self-oxidation should occur in the meantime of CO₂ reduction if O₂ evolution can not be observed. This self-oxidation issue would contradict with the claimed "long-term stability" even if the continuous flow is applied. Therefore, I could not recommend this manuscript for publication unless the authors can provide O₂ evolution results, which should be exactly correlated with quantity of CO₂ reduction product.

Version 1:

Reviewer comments:

Reviewer #2

(Remarks to the Author)

In this version, the authors have addressed all the comments. Now I think this manuscript could be accepted for publication.

Reviewer #3

(Remarks to the Author)

The authors have addressed my concern and the revision is acceptable.

Reviewer #4

(Remarks to the Author)

After careful evaluation, it is concluded that the manuscript presents several unique insights and methodological

advancements in addressing specific scientific questions. While the authors have provided relatively comprehensive responses to Reviewer 1's 6th, 7th, 8th, and 10th comments, their replies to the remaining comments still inadequate. The arbitration opinion is summarized as follows:

1. The proposed gas-liquid-solid three-phase interface construction strategy is not an original concept within the field. Numerous prior studies have demonstrated analogous three-phase interface electrodes in photocatalytic, photothermal catalytic, and photoelectrocatalytic applications (e.g., doi.org/10.1016/j.jcis.2025.01.187; doi.org/10.1016/j.ccllet.2025.112004; doi.org/10.1016/j.apsusc.2025.164650; doi.org/10.1038/s41467-025-60636-1). The role of such interfaces in enhancing mass transfer, enabling continuous flow operation, and improving catalytic performance has been extensively documented. Therefore, the manuscript must refine its scientific narrative to emphasize distinctive contributions that go beyond established knowledge.
2. The CO₂ photocatalytic conversion efficiency reported in this work is not competitive when benchmarked against leading-edge systems. Moreover, critical performance metrics, such as the CO₂ conversion rate, are absent. To clearly establish the technical significance of this study, the authors are strongly advised to include a comprehensive comparison table incorporating recently published benchmark catalysts under comparable reaction conditions.
3. The authors' attempt to draw parallels between electrochemical gas diffusion electrodes and the flow-enabled photocatalytic system to explain enhanced mass transfer is conceptually suggestive but lacks rigor. Although structural similarities exist in device design, the operational mechanisms differ significantly due to distinct driving forces and reaction environments. The innovative application of analytical titration principles to quantitatively assess the theoretical CO₂ mass transfer efficiency across different reactor configurations represents a methodologically sound advancement. This approach provides indirect evidence that the engineered three-phase interface substantially increases gas-liquid contact area compared to conventional configurations, thereby facilitating interfacial mass transfer. This methodological contribution holds value and broader implications for reactor engineering. However, to prevent misinterpretation, the authors should explicitly label the time point of each image in Figure S3.
4. In response to Reviewer 1's 4th comment, the authors have not provided a direct or satisfactory explanation. To substantiate their claims regarding flow-dependent performance, experimental data on CO₂ conversion rates under systematically varied gas flow rates must be included. Such data are essential for establishing a causal relationship between hydrodynamic conditions and catalytic output.
5. The discussion on the microscopic mechanism underlying the synergistic interaction between CO₂ and H₂O at the three-phase interface remains insufficient. Beyond macroscopic performance comparisons, the authors should provide mechanistic insights into microscale transport phenomena. Specifically, analysis of CO₂ and H₂O diffusion behavior and adsorption kinetics in porous media under different gas-liquid flow rates, supported by in-situ spectrum, electron microscope image or computational simulations, would strengthen the interpretation of dynamic interfacial processes.
6. Regarding Reviewer 1's 9th comment, observed differences in the Cu K-edge extended X-ray absorption fine structure (EXAFS) spectra between the flow-enabled and flow-less systems suggest potential structural distinction. The authors must conduct a rigorous assessment to determine whether these structural distinctions influence catalytic activity. It is imperative to strictly decouple material-level structural distinction from hydrodynamic effects to ensure that performance differences are only attributable to flow-induced mass transfer enhancement.
7. With respect to Reviewer 1's 10th comment, the presence of a C-C signal in XPS is commonly attributed to adventitious carbon contamination (amorphous carbon) originating from ambient exposure, which serves as the "internal standard" for peak position calibration and correction of all other elements. The authors' explanation is scientifically valid and fully consistent with standard practices in materials characterization.

Version 2:

Reviewer comments:

Reviewer #4

(Remarks to the Author)

The authors have provided comprehensive responses to all the comments. Following minor revisions addressing two specific points, the manuscript will be suitable for acceptance. The required revisions are as follows:

1. To enhance clarity for readers unfamiliar with the experimental setup, the authors should explicitly clarify the relationship among the X-axis variables across the three subplots in Figure S8, and explicitly state whether the reported pressure and flow rate values correspond to identical conditions on both sides of the catalytic membrane.
2. In Figure S1a, the color-coded particles represent distinct chemical species; the legend and caption must unambiguously identify CO₂, H₂O, and TiO₂. Additionally, Figure S1c is currently missing a legend and must be supplemented accordingly.

Responses to the Referees' Comments and the Corresponding Revisions

We extend our gratitude to the referees for their valuable feedback. After carefully considering their comments, we have made major revisions to the manuscript. We provide our responses to their comments below, along with the corresponding changes we have made to the main text and supplemental materials.

Response to Reviewer 1:

Comments 1. Please clarify the key advantages of the *flow-enabled* photocatalytic system developed herein over the existing flow systems (e.g., DOI: <https://doi.org/10.1002/anie.202302979>), specifically comparing mass transfer efficiency, resistance to carbon deposition, and reactive efficiency. Compared to your group's previous research on the "Continuous-flow reactor with superior production rate and stability for CO₂ reduction using semiconductor photocatalysts", which also utilized a flow system, the innovation of this paper is open to debate.

Response 1: We thank the reviewer for asking us to clarify what is new here. **Many reported "flow" photoreactors are effectively two-phase systems**, typically liquid-gas or gas-solid, that mainly improve light delivery and phase-specific mass transport¹⁻³; they can be viewed as **extensions of batch systems** (Supplementary Fig. 1a-b).

In contrast, our study runs the reaction as a GDE (gas-diffusion-electrode) based three-phase (gas-liquid-solid) interface (Supplementary Fig. 1c). This follows the GDE flow cell concept established in electrocatalysis for poorly soluble gases⁴. Just as the adoption of GDEs enabled a step beyond H-cell type two-phase operation, our system achieves a similar step-change under photocatalytic conditions. To the best of our knowledge, this has rarely been realized as a fully developed photocatalytic flow operation. We use CO₂ as a representative case, but the platform is general to other low-solubility gases (e.g., O₂, N₂).

Practically, in our system, H₂O flows over the photocatalyst layer, while CO₂ is fed from the back side through a hydrophobic gas diffusion layer (GDL). Gas reaches the photocatalyst through the porous backing; the liquid side remains wet but not flooded. By balancing the liquid flow and the gas-side back-pressure, the surface is continuously renewed, so reactants arrive easily and intermediates/products do not accumulate.

Here, we **focus on maximizing stability in the *flow-enabled* system and elucidating the underlying mechanisms of stability enhancement**. Building on our previous work that introduced the *flow-enabled* photocatalytic reactor concept⁵, **we now demonstrate why stability improves and how to overcome the prior lifetime limitation**. Using surface characterization and in-situ analyses, we show that *flow-less* operation accumulates carbon species on the surface, while three-phase flow keeps such deposits minimal. We also show that removing the accumulated carbon restores activity, confirming that surface carbon is a key deactivation factor that our flow operation suppresses. These observations are consistently supported by XPS/FT-IR/MALDI comparisons between *flow-enabled* and *flow-less* cases. As a result, we sustain photocatalytic operation for up to 15 days.

(New) **Supplementary Fig. R1** | Photocatalytic Reactor system configuration. **a**, Schematic illustrations of *flow-less* two-phase batch reactors; **b**, batch-modified flow reactors; and **c**, a gas diffusion electrode (GDE)-modified three-phase flow reactor. (left) Schematic illustration of the continuous *flow-enabled* photocatalytic reactor showing key components: gas pressure regulator, gas flow plate, mass flow meter (MFM), peristaltic pump, and PTFE membrane with photocatalyst coating. (right) Detailed gas diffused photocatalyst layer highlighting the role of the triple-phase interface in facilitating mass transport of CO₂ and H₂O.

To address this comment, we added new text and several new references in the Introduction.

(New Ref. 28) Li, Y. et al. A modular tubular flow system with replaceable photocatalyst membranes for scalable coupling and hydrogenation. *Angew. Chem. Int. Ed Engl.* **62**, e202302979 (2023).

(New Ref. 29) Yuan, Y. et al. Earth-abundant photocatalyst for H₂ generation from NH₃ with light-emitting diode illumination. *Science* **378**, 889–893 (2022).

(Ref. 11) Fang, S. et al. Photocatalytic CO₂ reduction. *Nat. Rev. Methods Primers* **3**, 1–21 (2023)

(Ref. 30) Jung, H. *et al.* Continuous-flow reactor with superior production rate and stability for CO₂ reduction using semiconductor photocatalysts. *Energy Environ. Sci.* **16**, 2869–2878 (2023).

(Main text) Previous strategies to enhance photocatalyst stability have largely focused on material modifications, including bandgap engineering, co-catalyst integration, and surface passivation. While these approaches have led to modest stability improvements, typical operational durations still remain below ~10 hours. Furthermore, they usually involve complex, multistep syntheses that hinder scalability and reproducibility.

From a reactor system-engineering perspective rather than further materials modification, many reported ‘flow’ photoreactors remain essentially two-phase, most commonly liquid-gas or gas-solid, where the primary gains are improved illumination and phase-specific mass transport^{11,28,29} (Supplementary Fig. 1a-b). However, these configurations typically cannot maintain simultaneous and continuous contact of both CO₂ gas and H₂O, as the hydrogen source, at the catalyst surface. Consequently, for poorly soluble gases like CO₂, challenges such as product accumulation and catalyst flooding can persist.

Recognizing these limitations, our group recently developed a three-phase (gas-liquid-solid) continuous-flow photocatalytic system, inspired by gas diffusion electrode technology from electrocatalysis³⁰ (Supplementary Fig. 1c). This system demonstrated significantly enhanced production rates and stability by maintaining simultaneous gas and liquid flow at opposite sides of the catalyst layer. However, the fundamental mechanisms underlying this remarkable stability improvement remained unclear.

~~In this study, we introduce a dynamic flow strategy that continuously supplies CO₂ and H₂O across the photocatalyst surface. This~~ In this study, we elucidate why the three-phase flow configuration achieves such superior stability. By comparing *flow-enabled* and *flow-less* conditions we reveal that this approach mitigates mass transport limitations, prevents the accumulation of products and intermediates, and inhibit the formation of deactivating surface species. Under optimized flow conditions, we demonstrate a significant enhancement in the long-term stability of widely used photocatalysts, including TiO₂, ZnO, CdS, and C₃N₄. Notably, TiO₂ maintained 80% of its initial activity over 15 days, surpassing previously reported stability metrics by more than two orders of magnitude. Our findings suggest that incorporating continuous flow conditions can serve as a foundational design principle for developing highly durable photocatalytic CO₂ reduction systems.

Comments 2. The paper uses CO production as the primary example, but the CO₂ reduction to CO reaction process is relatively simple. Is this flow strategy applicable to the production of complex hydrocarbon products with longer residence times and more complex reaction processes, such as ethanol and other multi-carbon products?

Response 2: Yes, in addition to CO production, our flow strategy is applicable to complex hydrocarbon products. We agree that our primary product is CO, which is an early-stage two-electron product in CO₂ reduction. Under continuous flow, the stable gas-liquid-solid (three-phase interface) contact, rapid desorption, and continuous sweep of surface species favor sustained

operation for such early pathways. However, **the formation of multi-electron and C₂⁺ products are also achievable in this reactor platform.**

Multi-electron and C₂⁺ products are achievable with appropriate photocatalyst materials^{6,7}. For C₂⁺ products (C₂H₄, C₂H₆), Cu-based catalysts such as CuO_x-polycrystalline ZnO⁸ and Cu₂O nanorods⁹ are particularly effective due to their C-C coupling ability. For CH₄ formation (8-electron reduction), catalysts like metal co-catalyst with semiconductor composite such as Pt-TiO₂, Pd-Nafion-TiO₂, and NiO_x/Ni have shown excellent selectivity. When such photocatalysts could be integrated into our continuous-flow reactor with optimized operating conditions, we expect higher production rates and improved stability due to the same interfacial control mechanisms.

To demonstrate this principle, among these, we tested a CH₄-selective photocatalyst (TiO₂-MXene composite) in our continuous-flow reactor to verify the applicability beyond simple CO production. These preliminary results are from our ongoing follow-up study. When integrated into our flow system with optimized conditions, we observed enhanced CH₄ formation rates and stability compared to the *flow-less* condition (**Supplementary Fig.6**). We achieved a ~6-fold increase in CH₄ production, with CH₄ selectivity maintained as the primary product (**Supplementary Fig.6a**). The stability improvement was even more pronounced for CH₄ production (**Supplementary Fig.6b**): while the batch system showed significant deactivation within a few hours, the *flow-enabled* system maintained stable CH₄ production for over 1500 minutes (>25 hours), demonstrating that our flow strategy effectively stabilizes complex multi-electron reactions. We believe C₂⁺ products would similarly benefit from the *flow-enabled* platform through optimization of flow parameters and surface residence time.

In summary, while our current work focuses on CO production for stability demonstration, the *flow-enabled* platform provides the necessary architectural flexibility and operational control to support more complex multi-carbon product formation. While comprehensive C₂⁺ product studies would require extensive optimization beyond the current scope focusing on stability demonstration, our CH₄ results and platform flexibility strongly suggest this is a viable direction for future work.

To address this comment, we added new **Supplementary Fig.6** and referred it in main text.

(New) **Supplementary Fig. 6** | Comparison of **a**, CH₄ production rate and **b**, stability with SPR of TiO₂-MXene composite photocatalyst under *flow-enabled* versus *flow-less* conditions. Error

bars in **a** represent standard deviations from two replicate experiments. All photocatalysts were tested under $100 \text{ mW}\cdot\text{cm}^{-2}$ illumination with a Xe lamp.

(*Main text*) As shown in **Fig. 2a and Supplementary Fig. 5**, the production rate in the *flow-less* system exhibits a sharp decline within a few hours, followed by saturation. A decrease below 20% of the initial performance was regarded as a loss of catalytic activity, consequently, no further measurements were conducted. Similarly, ZnO retained stability over extended periods (~100 hours), highlighting the effectiveness of flow conditions in stabilizing UV-light-driven photocatalytic materials (**Fig. 2b**). Notably, when testing alternative reaction pathways such as CH₄ production (8-electron pathway) using a TiO₂-MXene photocatalyst under identical *flow-enabled* conditions, the system similarly demonstrated enhanced performance and stability compared to *flow-less* batch systems (**Supplementary Fig. 6**), confirming that flow regulation enables diverse photocatalytic transformations beyond simple CO production.

Comments 3. While the author mentions that “Introducing a continuous flow of CO₂ and H₂O across the photocatalyst surface addresses these limitations by enhancing mass transport”, quantitative evidence for this mass transfer enhancement is lacking.

Response 3: Thank you for the constructive suggestion. To address this concern, we performed a reactor-intrinsic CO₂ uptake test under strong-sink conditions. This test provides **quantitative evidence demonstrating a 15-fold enhancement** in CO₂ mass transfer in our *flow-enabled* system.

Furthermore, our claim is grounded in the same transport physics in electrochemical CO₂ reduction reaction gas-diffusion and MEA reactors which directly translate to our *flow-enabled* photocatalytic geometry. The introduction of gas and liquid flow into GDE dramatically improves CO₂ transport to the catalyst compared to static, planar electrodes immersed in electrolyte.

Traditional H-cell configurations suffer from a thick CO₂ diffusion boundary layer (~100 μm) and low CO₂ solubility, limiting the current density to the order of ~10 mA cm⁻²¹⁰. In contrast, flow-cell designs with GDEs maintain a much thinner effective diffusion layer at the catalyst interface by directly feeding CO₂ gas to the catalyst through a porous backing. This raises the local CO₂ concentration at the reaction site and enables significantly higher reaction rates^{11,12}. Quantitatively, switching from a planar, liquid-phase CO₂ supply to a gas-fed GDE can raise the limiting current density by roughly one to two orders of magnitude.

In one study, a hydrophobic catalyst layer modification (adding PTFE to create gas pockets) doubled the CO₂RR partial current (to >250 mA cm⁻²) and single-pass conversion (~14%) versus a normal catalyst layer. The authors attributed this boost to a “balanced gas/liquid microenvironment that reduces the diffusion layer thickness, accelerates CO₂ mass transport, and increases CO₂ local concentration” near the catalyst. In summary, introducing gas flow or GDE architectures reduces the diffusion boundary layer and maintains higher CO₂ local concentration at the catalyst, thereby elevating the limiting current density by an order of magnitude or more¹³.

To quantitatively evidence transport enhancement imparted by our *flow-enabled* reactor, we performed a CO₂ uptake test under strong-sink conditions (0.10M NaOH; identical gas-liquid contact area and temperature in both *flow-enabled* and *flow-less* reactors without light and photocatalyst). Using the time-to-endpoint with phenolphthalein (flow of CO₂ over a fixed alkaline

load), the flow reactor reached the endpoint in 2 h with 10 mL NaOH solution, whereas the *flow-less* (batch headspace) cell required 3 h with 1 mL NaOH solution under the same geometric area ($A = 0.785 \text{ cm}^2$ of 1 cm diameter hole). Converting the consumed base to moles of CO_2 and normalizing by area and time gives:

The moles of CO_2 (n_{CO_2}) at endpoint,

$$n_{\text{CO}_2} \approx C_{\text{NaOH}} \times V_{\text{NaOH}} (\text{CO}_2 + \text{OH}^- \rightleftharpoons \text{HCO}_3^- \text{ at the phenolphthalein endpoint, pH} \approx 8.2)$$

To avoid bias from liquid-layer thickness, we report the area- and time-normalized CO_2 flux,

$$J = n_{\text{CO}_2}/(A \times t)$$

$$J_{\text{flow-enabled}} = C_{\text{NaOH}} \times V_{\text{NaOH}}/At = 1.0 \times 10^{-1} \times 1.0 \times 10^{-2} / (7.85 \times 10^{-5} \times 7200) = 1.76 \times 10^{-3} \text{ mol m}^{-2} \text{ s}^{-1}$$

$$J_{\text{flow-less}} = C_{\text{NaOH}} \times V_{\text{NaOH}}/At = 1.0 \times 10^{-1} \times 1.0 \times 10^{-3} / (7.85 \times 10^{-5} \times 10800) = 1.18 \times 10^{-4} \text{ mol m}^{-2} \text{ s}^{-1}$$

$$J_{\text{flow-enabled}}/J_{\text{flow-less}} \approx 15$$

The corresponding apparent permeance, $P = J / (y_{\text{CO}_2} \times p)$ (identical $y_{\text{CO}_2} = 1$ and $p = 1.2 \text{ bar}$ in both system)

$$P_{\text{flow-enabled}} = 1.47 \times 10^{-8} \text{ mol m}^{-2} \text{ s}^{-1} \text{ Pa}^{-1}$$

$$P_{\text{flow-less}} = 9.8 \times 10^{-10} \text{ mol m}^{-2} \text{ s}^{-1} \text{ Pa}^{-1}$$

$$P_{\text{flow-enabled}}/P_{\text{flow-less}} = J_{\text{flow-enabled}}/J_{\text{flow-less}} \approx 15$$

Despite using 10× less solution volume (1 mL vs. 10 mL), the *flow-less* system still required 50% more time (3 h vs. 2 h), demonstrating the dramatic mass transfer advantage of the flow configuration. Moreover, the flow cell decolorized uniformly across the interface, whereas the *flow-less* cell decolorized from the gas-liquid interface downward, leaving the bottom region colored for the longest time. This qualitative observation confirms non-uniform CO_2 delivery in the batch geometry. Time-lapse images (New **Supplementary Fig. 3**) clearly show this uniformity contrast under identical contact areas.

To address this comment, we have added the quantitative analysis of CO_2 mass transfer enhancement in the Supplementary Information (**Supplementary Fig. 3** and **Supplementary note 2**).

Supplementary Note 2: Quantitative Analysis of CO_2 Mass Transfer Enhancement

To quantitatively validate the mass transfer enhancement in our *flow-enabled* photocatalytic system, we conducted CO_2 uptake experiments under strong-sink conditions. This approach provides direct evidence of improved CO_2 transport compared to conventional *flow-less* batch systems.

CO₂ uptake rates were measured using 0.10 M NaOH solution as a strong CO₂ sink, with phenolphthalein as the endpoint indicator. Both *flow-enabled* and *flow-less* reactors were tested under identical conditions:

- Gas-liquid contact area: 0.785 cm² (1 cm diameter opening)
- Temperature: 25°C
- CO₂ pressure: 1.2 bar
- No photocatalyst or light irradiation

For the *flow-enabled* system, 10 mL of NaOH solution was circulated while CO₂ gas flowed at 10 sccm. For the *flow-less* system, 1 mL of NaOH solution was placed in a sealed chamber with CO₂ headspace. The endpoint was determined when the phenolphthalein indicator turned colorless (pH ≈ 8.2).

Results and Calculations

The CO₂ uptake was calculated based on the neutralization reaction:

1) *Flow-enabled* system:

Time to endpoint: 2 hours

NaOH volume: 10 mL

Moles of CO₂ absorbed: $n_{\text{CO}_2} \approx C_{\text{NaOH}} \times V_{\text{NaOH}} = 0.10 \text{ M} \times 0.010 \text{ L} = 1.0 \times 10^{-3} \text{ mol}$

2) *Flow-less* system:

Time to endpoint: 3 hours

NaOH volume: 1 mL

Moles of CO₂ absorbed: $n_{\text{CO}_2} = 0.10 \text{ M} \times 0.001 \text{ L} = 1.0 \times 10^{-4} \text{ mol}$

The area- and time-normalized CO₂ flux (J) was calculated as:

$$J = n_{\text{CO}_2} / (A \times t)$$

Where A is the gas-liquid contact area and t is the time to endpoint.

Flow-enabled: $J = (1.0 \times 10^{-3} \text{ mol}) / (7.85 \times 10^{-5} \text{ m}^2 \times 7200 \text{ s}) = 1.76 \times 10^{-3} \text{ mol m}^{-2} \text{ s}^{-1}$

Flow-less: $J = (1.0 \times 10^{-4} \text{ mol}) / (7.85 \times 10^{-5} \text{ m}^2 \times 10800 \text{ s}) = 1.18 \times 10^{-4} \text{ mol m}^{-2} \text{ s}^{-1}$

The apparent permeance (P) was calculated as:

$$P = J / (y_{\text{CO}_2} \times p)$$

Where $y_{\text{CO}_2} = 1$ (pure CO₂) and $p = 1.2$ bar for both systems.

Flow-enabled: $P = 1.47 \times 10^{-8} \text{ mol m}^{-2} \text{ s}^{-1} \text{ Pa}^{-1}$

Flow-less: $P = 9.8 \times 10^{-10} \text{ mol m}^{-2} \text{ s}^{-1} \text{ Pa}^{-1}$

Enhancement factor: $P_{\text{flow-enabled}} / P_{\text{flow-less}} = J_{\text{flow-enabled}} / J_{\text{flow-less}} \approx 15$

Conclusion

The quantitative CO₂ uptake experiments demonstrate that our *flow-enabled* photocatalytic system achieves a 15-fold enhancement in CO₂ mass transfer compared to conventional *flow-less* batch systems. This dramatic improvement in mass transport is a key factor contributing to the superior long-term stability and sustained activity observed in our *flow-enabled* photocatalytic CO₂ reduction system.

(New) **Supplementary Fig. 3 | Time-lapse images of the phenolphthalein decolorization during CO₂ exposure for the same geometric gas-liquid contact area.** **a**, The *flow-enabled* cell decolorizes uniformly across the interface. Inset: 0.10 M NaOH solution before and after the test. **b**, The *flow-less* (pressurized gas line in headspace of bottle) cell decolorizes from the gas-liquid interface downward, leaving the bottom region colored longest (yellow arrow), consistent with non-uniform CO₂ delivery and slower intra-liquid mixing in the batch geometry. Contact area = 1cm diameter $\approx 0.785 \text{ cm}^2$, without light irradiation and photocatalyst.

Also, we revised with these explanations into the main text

(Main text) Introducing a continuous flow of CO₂ and H₂O across the photocatalyst surface addresses these limitations by enhancing mass transport (**Fig. 1b** and **Supplementary Fig. 1-2 & Supplementary Note 1**). Quantitative analysis revealed a 15-fold enhancement in CO₂ mass transfer in our *flow-enabled* system compared to the *flow-less* configuration, as validated through CO₂ uptake experiments under strong-sink conditions (**Supplementary Fig. 3** and **Supplementary Note 2**). This dynamic flow facilitates the efficient removal of products and intermediates from the catalyst surface, preventing their accumulation and associated deactivation effects. The continuous supply of reactants ensures a consistent reaction environment, reducing the formation of deactivating surface species and maintaining catalyst activity over extended periods. By comparing these two systems, the advantages of continuous flow in enhancing photocatalytic performance and stability become evident, underscoring its potential for sustainable solar-to-chemical energy conversion applications.

Comments 4. Would the flow control strategy affect the CO₂ conversion while improving stability?

Response 4: Yes, the flow control strategy enhances both CO₂ conversion and stability simultaneously, and there is no trade-off. In our reactor, CO₂ and H₂O flow rates determine interfacial residence time and local CO₂ activity. The continuous CO₂ and H₂O flows remove surface-bound intermediates and poorly reactive products, preventing re-adsorption and site blocking. This keeps the gas-liquid-solid interface open to fresh reactants and suppresses secondary reactions that accumulate under the *flow-less* condition. This mechanism both increases conversion efficiency and extends operational lifetime. Our flow-dependence tests in **Fig.3** confirm this: reducing H₂O flow or CO₂ flow caused both conversion rates and stability to decline, proving that optimal flow conditions benefit both performance metrics through enhanced mass transport and surface renewal.

Comments 5. The authors attribute the outstanding catalytic activity in part to the formation of a stable gas-liquid-solid (three-phase) interface. In this work, how does a flow-enabled photocatalytic system dynamically regulate the interfacial balance among the gas, liquid, and solid phases? Please elaborate on the synergistic interaction mechanism between CO₂ and H₂O at the three-phase interface.

Response 5: Beyond improved CO₂ access, performance also depends on the microenvironment actively set by the flowing media. This flow balance governs activity, with three-phase control being the key determinant. When it is not properly established, performance decreases (**Supplementary Fig. S4**).

1. Dynamically regulation of the interfacial balance between catalyst layer

To optimize CO₂ and H₂O flow balance, we engineered a specialized photocatalytic flow reactor consisting of: gas flow plate, non-conductive GDL, photocatalyst layer, electrolyte flow plate, quartz window, and light source. All plates are fabricated from SUS, which remains inert under the photochemical conditions employed (new **Supplementary Fig. 1-2**).

Precise control of flow behavior, reactant pressure and flow rate, is essential for achieving high-performance photocatalytic reduction. Establishing a robust three-phase contact on the photocatalyst surface required iterative optimization; the practical solutions are summarized below.

Three design elements were critical. First, we defined a clear optical path through the H₂O transparent via a quartz window, ensuring that light reaches the photocatalyst surface to drive reactions at the three-phase interface. Second, we implemented precise gas delivery using a pressure regulator and flow control valve, while monitoring the flow rate with a mass flow meter (MFM). Third, we constructed a continuous-flow H₂O unit. To tune the balance between pressurized gas feed (set by pressure and flow rate) and H₂O circulation, the H₂O loop was configured as a closed-circulation pipeline connected on both sides of the photocatalyst layer. This adjustable, pressurized gas supply also increases the dissolution of reactants into the H₂O, thereby enhancing the production rate of the photocatalytic reaction.

2. Synergistic interaction mechanism between CO₂ and H₂O at the three-phase interface.

On the liquid side, continuous H₂O circulation clears accumulated species at the surface and reduces the stagnant layer, facilitating reactant transport. When the liquid flow is below the effective regime, purging is insufficient, local mass transfer slows, and CO re-adsorption increases, degrading overall kinetics. In contrast, overly high liquid flow raises hydraulic pressure and drives electrolyte into the porous layer (incipient flooding), progressively blocking gas pathways and lowering activity.

On the gas side, increasing CO₂ flow improves reactant delivery and gently strips gaseous products and weakly bound intermediates up to an optimum where residence time matches the surface adsorption–reaction–desorption timescale; beyond this window, residence becomes too short and the efficiency falls.

Gas pressure also shows an optimal window. Moderate pressurization maintains stable gas contact and enlarges the effective three-phase area. However, excessive pressure disturbs the interfacial balance and hinders product release by skewing the adsorption-desorption equilibrium.

In summary, balanced CO₂ and H₂O flows maintains the three-phase interface essential for sustained activity and selectivity. Deviation from optimal conditions accelerates deactivation.

In response to the editor's concern about operational details, we have added comprehensive reactor schematics and operating parameter guidelines in the Supplementary Information (**Supplementary Fig. 1-2** and **Supplementary note 1**)

(New) **Supplementary Fig. 2** | Photograph of the *flow-enabled* photocatalytic reactor system

Supplementary Note 1. Operational procedures for *flow-enabled* system

Prior to each experiment, the reactor was assembled and leak-tested. The photocatalyst-coated gas diffusion layer (GDL) was positioned with the catalyst side facing the H₂O flow plate, ensuring proper orientation for triple-phase interface formation. Start-up procedure involved first establishing H₂O circulation at 10 mL min⁻¹ to wet the catalyst surface, followed by gradual introduction of CO₂ flow from 1 to 10 sccm over 5 minutes while monitoring the system pressure (maintained at 1.2 bar). This gradual ramp-up prevents sudden pressure changes that could disrupt the gas-liquid interface or cause flooding. During operation, the balance between gas and liquid phases was monitored by observing bubbles and flooding.

Also, we revised with these explanations into the main text

(*Main text*) Introducing a continuous flow of CO₂ and H₂O across the photocatalyst surface addresses these limitations by enhancing mass transport (**Fig. 1b** and **Supplementary Fig. 1-2 & Supplementary Note 1**). Quantitative analysis revealed a 15-fold enhancement in CO₂ mass transfer in our *flow-enabled* system compared to the *flow-less* configuration, as validated through CO₂ uptake experiments under strong-sink conditions

Comments 6. Good stability is one of the basic conditions for industrial application. Does the flow strategy result in economically viable energy consumption and product separation costs in long-term operation, or does it sacrifice one for the other? It is recommended to supplement the discussion with a technical-economic analysis.

Response 6: We appreciate the suggestion and have conducted a simple techno-economic analysis (TEA) comparing batch and flow strategies under a common set of assumptions and a 1 kg-CO/day production target. The analysis shows costs of (i) reactor, (ii) catalyst, and (iii) product separation (PSA) contributions, allowing us to test whether flow operation requires a trade-off between energy and separation¹⁴.

We find that the flow strategy delivers substantially lower energy use and lower cost than the batch alternative, owing to its higher productivity and longer stability. By contrast, the batch configuration with ~50 times lower rate and only ~2 h stability demands orders of magnitude larger irradiated area and frequent catalyst replacement, yielding a much higher cost of CO. Moreover, if more active photocatalysts are employed, the flow strategy has the potential to become an economically viable CO₂ conversion system.

Importantly, in the flow system the CO product fraction can be readily tuned to ~10% which is appropriate for PSA by adjusting feed/purge rates. However, in the batch system, achieving such concentrations is impractical due to the low productivity. Detailed TEA boundary conditions and core results are provided below and we added our TEA results in new **Supplementary Note 3** and **Supplementary Table 2**.

Supplementary Note 3: TEA boundary conditions and assumptions¹⁻⁴:

- Target: 1 kg CO/day production (35.70 mol/day).
- Catalyst: TiO₂
- Catalyst productivity: flow system = 800 μmol/g_cat/h; batch system = 1/50 of flow cell.
- Optical utilization (irradiation efficiency): identical for both (normalized comparison).
- Catalyst loading: 1 mg/cm² (= 10 g/m²).
- Catalyst lifetime: flow 350 h, batch 2 h.
- Catalyst price: \$3.19/kg (<https://www.chemanalyst.com/Pricing-data/titanium-dioxide-52>)
- Reactor CAPEX: flow system \$905/m², batch system \$200/m²; project life 10year; discount 8% (CRF = 0.149)
- Electricity for PSA separation: (0.25 kWh/Nm³) x (feed volumetric flow rate) x (time for 1 kg CO production) x (\$0.03/kWh), Baseline cost scales inversely with CO mole fraction in the feed.
- PSA CAPEX: NREL PSA CAPEX scaling $C_{PSA} = 10.8M \times (\text{mole flow rate}/4.40 \times 10^6)^{0.65}$
Annualized CAPEX = $C_{PSA} \times \text{CRF}$
- CO mole fraction for PSA separation: 10% for both system
In flow system, CO concentration could be reached to 10% with 5,560 SCCM flow rate
In batch system, CO concentration could be reached to 10% with 65μm height cell, 2h operation

TEA core results:

Metric	Flow-enabled system	Flow-less system
Catalyst required	1.86 kg	92.98 kg
Irradiation area	186 m ²	9,298 m ²
Initial reactor CAPEX (@flow = \$905/m ² , batch = \$200/m ²)	\$168,300	\$1,860,000
Annualized CAPEX	\$68.7/kg-CO	\$759.1/kg-CO
Annual catalyst makeup	46.5kg/year	407,231kg/yer
Catalyst cost (@\$3.19/kg)	\$0.41/kg-CO	\$3,559.1/kg-CO
PSA Electricity (@\$0.25 kWh/Nm ³)	\$0.06/kg-CO	\$0.06/kg-CO
PSA CAPEX (NREL, CRF 0.149)	\$1.225/kg-CO	\$1.225/kg-CO
PSA Total (Electricity + CAPEX)	\$1.285/kg-CO	\$1.285/kg-CO
Total (Levelized cost of CO)	\$70.41/kg-CO	\$4,319.7/kg-CO

(New) Supplementary Table 2. Techno-economic comparison of *flow-enabled* and *flow-less* batch photocatalytic systems for CO₂ reduction

(*New Supplementary Ref. 1*) Jouny, M., Luc, W. & Jiao, F. General techno-economic analysis of CO₂ electrolysis systems. *Ind. Eng. Chem. Res.* **57**, 2165–2177 (2018).

(*New Supplementary Ref. 2*) Li, D. et al. Shaping and doping metal-organic framework-derived TiO₂ to steer the selectivity of photocatalytic CO₂ reduction toward CH₄. *Inorg. Chem.* **63**, 15398–15408 (2024).

(*New Supplementary Ref. 3*) National Renewable Energy Laboratory (NREL), About the carbon dioxide utilization data and analysis. <https://www.nrel.gov/bioenergy/co2-utilization-economics/about> (accessed 9 September 2025).

(*New Supplementary Ref. 4*) Spasiano, D., Marotta, R., Malato, S., Fernandez-Ibañez, P. & Di Somma, I. Solar photocatalysis: Materials, reactors, some commercial, and pre-industrialized applications. *A comprehensive approach. Appl. Catal. B* 170–171, 90–123 (2015).

Also, we added the following explanations text to main text.

(*Main text*) To evaluate the practical viability of our *flow-enabled* system, we conducted a techno-economic analysis comparing *flow-enabled* and *flow-less* strategies for a 1 kg CO/day production target (**Supplementary Note 3** and **Supplementary Table 2**). The analysis reveals that the flow system delivers substantially lower costs due to its 50-fold higher productivity and extended operational stability (360 h vs. 2 h), which dramatically reduces both irradiated area

requirements and catalyst replacement frequency. Furthermore, the flow configuration enables tunable CO concentrations (~10%) suitable for pressure swing adsorption (PSA), whereas achieving such concentrations in batch systems is impractical due to low productivity. These economic advantages, combined with the demonstrated stability improvements, position the *flow-enabled* photocatalytic system as a promising pathway toward industrial-scale CO₂ conversion.

Comments 7. The source of CO production activity needs verification.

Response 7: We appreciate the reviewer's important point about verifying the carbon source. We have previously performed rigorous isotope-labeling experiments to confirm the origin of CO in our *flow-enabled* reactor system (Jung et al., *Energy Environ. Sci.* 16, 2869–2878, 2023)⁵. Using the identical reactor configuration and photocatalyst (TiO₂) employed in this study, we conducted ¹³CO₂ labeling experiments. The mass spectrometry results clearly demonstrated that: CO signal shifted from m/z = 28 (¹²CO) to m/z = 29 (¹³CO) when switching from natural CO₂ to ¹³CO₂ - CH₄ signal correspondingly shifted from m/z = 16 to m/z = 17. No CO production was observed in control experiments without light or catalyst. These results unambiguously confirm that CO originates from photocatalytic reduction of the supplied CO₂ not from any carbon contamination or decomposition of organic components.

To address this comment, we added the following text to the Result section

(*Ref. 30*) Jung, H. *et al.* Continuous-flow reactor with superior production rate and stability for CO₂ reduction using semiconductor photocatalysts. *Energy Environ. Sci.* **16**, 2869–2878 (2023).

(*Main text*) Remarkably, the *flow-enabled* platform demonstrated significantly enhanced stability. For instance, TiO₂ retained over 80% of its initial activity for more than 360 hours (>15 days), achieving cumulative product formation rates up to 825 μmol·g⁻¹·h⁻¹—far exceeding the performance observed under batch conditions without CO₂ and H₂O flow. The carbon source of CO production in this *flow-enabled* reactor configuration has been verified through ¹³CO₂ isotope labeling experiments in our previous work³⁰, confirming that CO originates from photocatalytic CO₂ reduction. This represents a more than two order of magnitude enhancement in stability compared to previously reported TiO₂-based photocatalysts^{21,37–53}, showing the potential of flow-regulated operation in overcoming longstanding barriers to practical photocatalytic CO₂ reduction.

Comments 8. Although XPS/FTIR/MALDI-TOF have demonstrated that carbon species accumulation is the primary cause of deactivation, there is a lack of quantitative models or theoretical calculations to support the microscopic mechanisms of how flow dynamically inhibits carbon deposition, such as interfacial mass transfer kinetics and adsorption-desorption equilibrium. If DFT calculations or microkinetic simulations could be supplemented, this would significantly enhance the persuasiveness of the mechanism.

Response 8: We appreciate the reviewer's thoughtful suggestion regarding quantitative modeling. We fully agree that atomistic DFT or microkinetic simulations would deepen the microscopic

picture. While the dynamic effects of flow on carbon deposition are still not fully understood at the molecular level, such **theoretical calculations present significant challenges due to the complex interplay between fluid dynamics, mass transport, surface reactions, and time-dependent carbon accumulation processes that occur over vastly different length and time scales.** The difficulty in directly simulating flow effects in photocatalytic systems arises from several factors: (i) the need to couple continuum-level transport with atomistic surface processes, (ii) the multi-phase nature of the gas-liquid-solid interface, and (iii) the long timescales (hours to days) over which carbon accumulation occurs. These complexities make direct simulation computationally prohibitive with current methods.

To place these observations in a theoretical context without overextending the scope, we reference representative continuum and microkinetic studies in electro-GDE/flow-cell CO₂ conversion that formalize how convection and product removal control interfacial mass transfer and adsorption-desorption equilibria. Continuum models show that gas-phase delivery and flow thin the concentration boundary layer, set spatially non-uniform active zones, and impose a trade-off between flow rate and single-pass conversion¹⁵. On the coverage side, DFT-informed modeling shows that in an open gas-feed environment (**high CO₂ flux**), **rapid CO desorption** limits hydrogenation and C-C coupling, whereas confinement increases CO residence/accumulation and promotes C-C coupling¹⁶. In gas-fed GDE flow cells, decreasing the CO₂ feed flow rate increases C₂⁺ selectivity at a fixed current density, while CO₂ availability remains sufficient; this trend has been attributed in part to a higher probability of CO re-adsorption and subsequent dimerization arising from longer residence and elevated interfacial CO¹⁷. In gas-fed CO₂ electrocatalysis, it has been widely reported that **CO generated from CO₂ can re-adsorb on Cu surfaces and undergo further reactions such as C-C coupling** to C₂⁺ products, with coverage and residence being governed by gas flux/flow and interfacial mass transfer^{18,19}. In addition, polymerization-type carbonaceous films arising from minor products/intermediates have been implicated in durability losses under high-rate operation^{20,21}, consistent with our observation that suppressing surface carbon deposition restores long-term stability.

These computational studies from electrochemical flow systems, while not directly applicable to our photocatalytic system, suggest that flow suppresses carbon buildup through similar mass transport mechanisms. Although the exact molecular-level mechanisms of how flow prevents carbon deposition remain not fully understood, the agreement between these **theoretical predictions and our experimental results supports our proposed explanation.** Developing complete simulations for photocatalytic flow systems remains computationally challenging due to the need to model multiple length and time scales simultaneously. Nevertheless, the principles learned from electrochemical systems provide valuable guidance for future theoretical work aimed at further optimizing flow conditions and extending operational lifetimes.

Comments 9. Are the catalysts used in Figures 4c and 4e the same? If so, why do the spectra at 0 h (before the reaction) show discrepancies in the relative intensities of certain peaks?

Response 9: Yes, the CuO catalysts in Fig. 4c and Fig. 4e are the same material and were prepared using the same loading rate and deposition protocol. **The slight difference in the peak intensity of the two as-prepared samples likely originates from practical variations during electrode preparation.** Since each electrode was freshly prepared with a relatively small catalyst loading, minor differences in sample amount and distribution are inevitable⁴⁰, which in turn affect

the electrode quality. These differences lead to higher noise levels in k-space (particularly above $\sim 8 \text{ \AA}^{-1}$ in **Supplementary Fig. 22**), and consequently cause small variations in the Fourier-transformed spectra. Importantly, the observed intensity difference is most noticeable in the second and third coordination shells, while no significant discrepancies are present in the overall spectral features. As our main focus was to follow the structural evolution of the catalyst from the as-prepared state under in-situ conditions, this small variation does not affect the validity of our conclusions. We added Cu K-edge XANES k-space of CuO from *flow-enabled* and *flow-less* system before reaction results in new **Supplementary Fig. 22**.

(New) **Supplementary Fig. 22** | Cu K-edge XANES k-space of CuO from *flow-enabled* and *flow-less* system before reaction

Also, we added the following explanations text to main text.

(Main text) CuO exhibits coordination changes and redox behavior under reaction conditions, which are effectively tracked by XAFS analysis^{74–76}. These characteristics make CuO a suitable model system for evaluating how flow conditions influence not only catalytic performance but also the structural integrity and active state of the photocatalyst in real time (**Fig. 4**). Both CuO samples were prepared identically, with minor variations attributed to differences in electrode preparation, as confirmed by identical pre-reaction Cu K-edge XANES spectra (**Supplementary Fig. 22**)

Comments 10. The study proposes that the flow strategy enhances catalyst stability by removing carbonaceous deposits. However, XPS results show the presence of C–C bonds on the catalyst surface even after prolonged reaction. Could the authors clarify how the flow system selectively removes specific types of carbon species?

Response 10: We thank the reviewer for this insightful comment. We clarify that our claim is not that the flow environment eliminates all surface carbon, but that it suppresses the accumulation of

deactivating, polymeric/condensed carbon by continuously removing weakly bound intermediates and products at the gas–liquid–solid interface.

Consistent with this view, the C–C signal detected by XPS predominantly originates from the PTFE/PFSA support rather than newly formed deposits^{41,42}. Quantitatively, the C-C/C-F₂ ratio of the reference (bare PTFE / as-prepared catalyst) is ~0.73 (**Supplementary Fig. 17**), and after prolonged operation under flow it remains 0.72, within the reference range. By contrast, under *flow-less* conditions the ratio increases to 1.71, accompanied by attenuation of C-F₂, which is consistent with formation of a hydrocarbon overlayer.

Mechanistically, the flow system shortens the residence time of reactive surface intermediates and enhances convective/desorptive removal, thereby preventing secondary coupling/condensation that would otherwise yield heavy, deactivating carbon under stagnant conditions.

To address this comment, we have added the C 1s XPS spectrum of bare PTFE film as **Supplementary Fig. 17** and revised the main text accordingly.

(New) **Supplementary Fig. 17** | C 1s XPS spectrum of bare PTFE film.

(*Main text*) XPS analysis of the C1S bonds was conducted to further investigate these surface changes (**Fig. 3c-e**). The reference photocatalyst layer (**Fig. 3c**) exhibited dominant C-F bonds from the PTFE GDL substrate, along with minor C-C, C=O, and O-C-O bonds⁶⁵ (the C-C/C-F₂ ratio of the reference and bare PTFE is ~0.73 in **Supplementary Fig.17**). In the *flow-enabled* sample (**Fig. 3d**), the C-F₂ and C-C peaks were similar to those of the reference (the C-C/C-F₂ ratio ~0.72), indicating minimal surface modifications. However, the *flow-less* system sample (**Fig. 3e**) showed a significant increase in the C-C bond intensity, surpassing the C-F₂ bond (the

C-C/C-F₂ ratio ~ 1.71). This result suggests that carbon-carbon bonds dominated the surface, likely masking the C-F₂ signal from the PTFE substrate due to carbonaceous species accumulating on the catalyst surface.

Comments 11. The performance enhancement in the flow-enabled photocatalytic system is attributed to continuous reactant supply and efficient product removal. However, the mass transfer efficiency has not been quantitatively analyzed in the manuscript.

Response 11: We thank the reviewer for this comment. We note that this point regarding quantitative mass transfer analysis closely aligns with Comment 3, which we have addressed comprehensively above.

To briefly summarize our response:

1. Quantitative mass transfer measurements through CO₂ uptake experiments showing a 15-fold enhancement in CO₂ flux:
Flow-enabled: $J = 1.76 \times 10^{-3} \text{ mol m}^{-2} \text{ s}^{-1}$
Flow-less: $J = 1.18 \times 10^{-4} \text{ mol m}^{-2} \text{ s}^{-1}$
2. Permeance calculations demonstrating the same 15-fold improvement in apparent mass transfer coefficient
3. Visual confirmation via time-lapse imaging showing uniform vs. non-uniform CO₂ delivery (**Supplementary Fig. 3**)

We believe Response 3 fully addresses the quantitative mass transfer analysis requested here. To avoid redundancy, we respectfully direct the reviewer to that detailed response and the corresponding additions to the manuscript.

Comments 12. The authors should review and improve the writing conventions, such as the inconsistent formatting of number subscripts in the abstract.

Response 12: Thank you for pointing this out. We have carefully reviewed the entire manuscript to fix inconsistent formatting. We corrected the chemical formula subscripts (e.g., CO₂, H₂O) and standardized the notation for units and symbols throughout the abstract, main text, and figures. All formatting now follows the journal's style guidelines consistently.

Response to Reviewer 2

Comments 1. Line 56: This approach mitigates mass transport limitations, prevents the accumulation of products and intermediates, and inhibits the formation of deactivating surface species. During the reaction process, what effects do the reactants CO₂ and H₂O have on the catalyst? How could the authors alleviate the effects of photo-corrosion under light exposure on catalysts?

Response 1: We appreciate the reviewer's insightful questions regarding the mechanistic roles of CO₂ and H₂O flows and photo-corrosion mitigation strategies.

1) Specific effects of CO₂ and H₂O on catalyst performance:

Our flow-variation study (**Fig. 3a-b**) indicates complementary roles. Continuous **CO₂ delivery sustains long-term activity by mitigating local accumulation of products/intermediates**, thereby reducing re-adsorption and undesired side reactions observed in the *flow-less* control. Continuous **H₂O circulation helps establish and maintain a stable gas-liquid-solid (triple-phase) interface at the GDL and removes byproduct species, reducing boundary layer thickness**. Consistently, in the *flow-enabled* configuration we observe sustained stability (SPR \geq 0.8) over 15 days for CO production (**Fig. 2e**), whereas partial or absent flows lead to faster decay (**Fig. 3a**). Surface analyses (XPS/FTIR/MALDI-TOF) corroborate this picture: *flow-less* operation yields carbonaceous residues on the photocatalyst surface, while *flow-enabled* samples remain close to the reference (**Fig. 3c-g**).

2) Addressing photo-corrosion under illumination:

Our approach does not claim to remove intrinsic material instabilities; rather, it **alleviates secondary, accumulation-mediated routes** that accelerate light-induced degradation. By continuously renewing reactants and removing weakly bound products/intermediates, **the flow operation reduces surface coverage by carbonaceous residues and associated deactivation pathways** (**Fig.3-4**). We acknowledge that catalysts intrinsically susceptible to photo-corrosion, e.g., CdS and CuO, will still degrade; indeed, while short-term operation showed no clear signatures of photo-corrosion, under extended operation revealed gradual activity decline that may be attributed to photo-corrosion - with CdS and CuO losing significant activity within \sim 80 h and \sim 40 h respectively, in contrast to the more robust TiO₂ system. Thus, the flow strategy extends operational stability without altering the material's intrinsic photo-stability.

Comments 2. Line 103: The product formation rate is up to 825 $\mu\text{mol}\cdot\text{g}^{-1}\cdot\text{h}^{-1}$ far exceeding the performance observed under batch conditions without CO₂ and H₂O flow. This represents a more than two order of magnitude enhancement in stability compared to previously reported TiO₂-based photocatalysts. In the flow-enabled platform, what is the pressure exerted by CO₂ and H₂O? Is the superior performance due to more CO₂ with higher local concentrations being involved in the reaction?

Response 2: Pressures were matched between the *flow-enabled* and *flow-less* systems, and the improved performance is best explained by convective mass transport instead of an increased local CO₂ concentration.

To clarify, both the *flow-enabled* and *flow-less* systems were operated under comparable pressure conditions. Specifically, a mild back pressure was maintained by a downstream needle valve in the *flow-enabled* system, while the batch system was pressurized to the same level before sealing the reactor. Therefore, the superior stability and productivity observed in the *flow-enabled* platform cannot be attributed to elevated CO₂ pressure or increased bulk CO₂ concentration.

Instead, the dramatic performance enhancement stems from the fundamental difference in mass transport mechanisms. The continuous flow creates convective transport that: (i) constantly renews the gas-liquid-solid triple-phase interface, (ii) maintains steady-state reactant concentrations at the catalyst surface, and (iii) rapidly removes both gaseous products and weakly adsorbed intermediates. This dynamic environment prevents the re-adsorption of products and the accumulation of deactivating carbonaceous species, as evidenced by our surface characterization data (**Fig. 3**).

In contrast, the *flow-less* system, despite having identical initial pressure and CO₂ concentration, suffers from diffusion-limited transport that leads to local product accumulation and subsequent catalyst poisoning. Thus, the observed improvements are definitively flow-driven rather than pressure-driven phenomena.

To address this comment, we added the following text to the caption of **Fig. 2**:

(Fig. 2) Both systems operated under identical pressure conditions, confirming that the enhanced stability arises from convective mass transport rather than pressure effects.

Comments 3. In this work, the authors optimized the flow parameter to predominantly generate CO as the primary product

What are the respective single-pass CO₂ conversion rates in the flow-less batch and flow-enabled platform?

Response 3: The single-pass conversion (SPC) is a flow-reactor metric defined as the fraction of the incoming CO₂ that is converted during one pass through the reactor. A closed, *flow-less* batch reactor does not feature a continuously renewed feed; therefore, SPC is not a well-defined metric for batch operation. For batch, we instead report production rate and stability (SPR), consistent with the main text.

For transparency, we add the time-dependent headspace CO mole fraction (ppm) for the closed batch system (**Supplementary Fig. 8**). Under the representative gas phase *flow-less* batch condition used in this study: 1 mg photocatalyst on a 1 cm² area in a 50 mL closed volume, and the headspace CO reached ~2 ppm at 1 h and gradually approached a plateau over several hours (**Supplementary Fig. 5a** and **Fig. 8a**). By contrast, in the *flow-enabled* system under the representative operating point (same catalyst condition; CO₂ = 10 sccm & 1.2 bar; H₂O flow = 10 mL/min), the per-pass CO₂ conversion is SPC $\approx 1.0 \times 10^{-6}$ (dimensionless; \approx “1 ppm-equivalent” in the effluent under a pure-CO₂ feed).

(New) **Supplementary Fig. 8** | Comparison of CO production for the *flow-less* and *flow-enabled* systems. **a**, time-dependent headspace CO mole fraction (ppm). **b**, accumulated CO production per 1 g of photocatalyst in each reactor.

Although the absolute value is modest at laboratory scale, SPC is a design-tunable parameter: increasing catalyst mass/active area or gas residence time (e.g., by lowering the CO₂ flow) raises SPC nearly linearly until transport limitations or incipient flooding arise (**Supplementary Fig. 9**). Consistent with this, **Supplementary Fig. 9a** shows that decreasing the CO₂ flow increases the steady-state effluent CO level, and **Supplementary Fig. 9b** shows a monotonic increase with catalyst loading. We observed an ~8-fold increase in the effluent product level under otherwise comparable conditions (**Supplementary Fig. 9c**) in a larger-aperture reactor (4 cm diameter, versus the 1 cm illuminated aperture used as standard shown as **Supplementary Fig. 9d**). At the process level, SPC can be further increased by introducing CO₂ recycle and/or a cascaded flow-through train⁴³, which we note as practical scale-up handles to be optimized in future studies.

(New) **Supplementary Fig. 9 | Tunability of single-pass CO₂ conversion (SPC) in the flow-enabled platform.** Outlet concentration as a function of **a**, gas flow rate and **b**, catalyst loading rate. **c**, Effect of reactor illuminated area: comparison between the standard 1 cm diameter aperture and a 4 cm diameter large system. **d**, Photograph of the standard and large size *flow-enabled* system. All photocatalysts were tested under 300 mW·cm⁻² illumination with a Xe lamp.

To address this comment, we have added new **Supplementary Fig. 8** and **9**, and revised the main text accordingly.

(*Main text*) To demonstrate the exceptional long-term stability achieved by our *flow-enabled* system, we conducted a 25-day continuous CO₂ reduction experiment using TiO₂, which exhibited the best stability and activity in our system under optimized flow conditions of 10sccm CO₂ gas and 10mL/min H₂O (**Fig. 2e**). In this continuous *flow-enabled* photocatalytic system, each flow parameter significantly influences both the production rate and selectivity of the CO₂ reduction reaction^{30,55}. In this work, we optimized the flow parameter to predominantly generate CO as the primary product (**Supplementary Fig. 7**). During the first two weeks, the system maintained its initial activity; however, by the 15th day, activity declined to approximately 80% of its initial level, followed by a further decrease thereafter. Notably, to the best of our knowledge, this level of photocatalytic stability represents the longest continuous

operation reported for CO₂ reduction using TiO₂, surpassing previously reported TiO₂-based systems by more than two orders of magnitude ²¹. From a practical continuous-production perspective, over equivalent operation periods the accumulated CO output in the *flow-enabled* TiO₂ reactor substantially exceeds that of the *flow-less* batch system (**Supplementary Fig. 8**). Furthermore, performance is design-tunable and can be fine-tuned through rational cell-geometry design and flow-rate optimization (**Supplementary Fig. 9**), providing additional avenues to enhance both activity and selectivity.

Comments 4. To enhance readability, please include schematic diagrams and actual photographs of the flow-enabled platform.

Response 4: We thank the reviewer for the helpful suggestion. In line with the format adopted in our response to Reviewer 1 (Comment 5), we have added (i) a concise schematic of the *flow-enabled* reactor in **Supplementary Fig. 1c** highlighting the continuous CO₂ and H₂O flow paths, the photocatalyst layer on PTFE film, sus housing and gasket with a 1 cm diameter aperture (illuminated area), and the illumination geometry. In **Supplementary Fig. 2**, (ii) annotated photographs of the assembled device with labeled inlets/outlets of gas and liquid flows.

Comments 5. What is the temperature of the reaction system? How can the condensation of incoming H₂O vapor into water be prevented?

Line 221-223: It is likely that H₂O flow has a greater influence on the initial reaction activity.

If it only affects the initial reaction, why maintain a continuous flow of H₂O at 10 mL/min into the reaction system?

Response 5: We appreciate the reviewer's detailed questions.

1) Operating temperature:

We directly measured the internal temperature in this study by inserting a thermocouple into the liquid plate adjacent to the illuminated window under the standard operating condition (Xe lamp, 300 mW cm⁻²; CO₂ 10 sccm; H₂O 10 mL min⁻¹). The reactor operated near ambient; over a 7-hour run the internal temperature remained stable at ~32 °C, and we added reactor temperature during *flow-enabled* operation in the Supplementary Information (**Supplementary Fig. 4**).

(New) **Supplementary Fig. 4 | Internal reactor temperature during *flow-enabled* operation.**

Also, we revised with these explanations into the main text

(*Main text*) **Figure 2** illustrates the broad applicability and exceptional long-term stability of *flow-enabled* photocatalytic approach across various materials, including TiO₂, ZnO, C₃N₄, and CdS, over 100-hour reaction times. Without any modifications to the photocatalysts, the CO₂ reduction activity of each material was evaluated under both continuous *flow-enabled* and traditional *flow-less* gas-phase batch system. The internal temperature of the reactor, measured via thermocouple insertion into the liquid plate, remained stable at ~32 °C throughout the 7-hour operation under standard conditions (**Supplementary Fig. 4**).

2) Prevention of condensation of incoming H₂O vapor into liquid water:

In the *flow-enabled* configuration, a circulating H₂O layer resides between the illuminated window and the catalyst support; under illumination, local heating can generate minor H₂O vapor. Importantly, we do not introduce H₂O as vapor into the CO₂ feed. Any vapor arises locally from the window-side water layer. Although the gas and liquid manifolds are physically isolated by the hydrophobic PTFE film, trace vapor could migrate toward the gas side through pores in the PTFE or via perimeter microgaps if the pressure balance is not well controlled. To preclude incidental moisture carryover and condensation, we maintain a small positive gas pressure, keep the CO₂ manifold proximal to the reactor isothermal and above the dew point, reinforce perimeter sealing, and place a short in-line dryer immediately upstream of the MFM (**Supplementary Fig. 2**). Under these controls, we did not observe uncontrolled condensation in the gas line.

3) Why maintain continuous H₂O flow (10 mL min⁻¹) if its effect is “initial”?:

While H₂O flow most critically impacts the initial phase, it remains essential throughout operation. Without active H₂O circulation, water infiltrates the GDE and compromises triple-phase interface

formation, causing activity to decline and plateau at a lower level (**Fig. 3a**). Continuous H₂O flow prevents this infiltration and maintains the interface integrity. After initial stabilization, CO₂ flow becomes the dominant factor for long-term performance, ensuring reactant supply and product removal, but H₂O flow continues its supporting role in preserving optimal reaction conditions.

Comments 6. To further validate the catalyst stability after the reaction, please offer the TEM and HRTEM images of the (used?) catalyst.

Response 6: We appreciate the reviewer's suggestion. We have added paired TEM/HRTEM images of pristine TiO₂ and the used TiO₂ after long-term operation in the *flow-enabled* platform (new **Supplementary Fig. 15**). The overview TEM panels show comparable primary particle morphology before and after reaction. In the corresponding HRTEM panels, clear lattice fringes are retained and the measured interplanar spacings remain at $d \approx 0.35$ – 0.36 nm, indexed to anatase TiO₂ (101). Within the spatial resolution of our measurements, no discernible loss of crystallinity or amorphization is observed, indicating that the TiO₂ lattice is preserved during extended operation.

To address this, we included a new figure that clearly shows stability of photocatalyst after long term reaction.

(New) **Supplementary Fig. 15 | TEM/HRTEM of TiO₂ before and after long-term *flow-enabled* operation. Overview of TEM image of a Pristine TiO₂ and b the TiO₂ after long-term operation. White boxes mark regions magnified in c and d, respectively. HRTEM images showing clear lattice fringes with interplanar spacings of $d \approx 0.35$ – 0.36 nm, indexed to anatase (101), indicating preserved crystallinity after reaction.**

We added text to the main text to refer to **Supplementary Fig. 15**.

(Main text) Scanning electron microscopy (SEM) and **high-resolution transmission electron microscopy (HRTEM)** of the catalyst layer before and after the reaction (**Supplementary Fig. 14 and 15**) revealed no significant changes in the loading or structure of the catalyst, indicating that the decline in activity is not caused by physical detachment of the catalyst but rather by surface modification phenomena or other factors affecting the catalyst's reactivity.

Comments 7. For in-situ X-ray absorption spectroscopy, the CuO is used as the model catalyst. So what are the products when CuO is used as the photocatalyst in the flow-enabled platform? Is it CO? Please provide supplementary information.

Response 7: Under the standard operating conditions of our *flow-enabled* platform (Xe lamp 100 mW cm⁻²; CO₂ 10 sccm; H₂O 10 mL min⁻¹), **CuO produced CO** as the dominant gaseous product. We have added the corresponding dataset to the Supplementary Information (**Supplementary Fig. 23**), reporting 5 h stability profiles and the CO production rate for CuO under *flow-enabled* and batch operation.

(New) **Supplementary Fig. 23 | Performance of CuO under *flow-enabled* and batch operation.** **a**, Stability over a 100 h reaction. **b**, CO production rate at 5 h. Reaction conditions: Xe lamp 100 mW cm⁻²; CO₂ 10 sccm; H₂O 10 mL min⁻¹.

Response to Reviewer 3

Comments 1. In the present study, the authors introduced the continuous CO₂ and H₂O flow during the photocatalytic CO₂ reduction process and achieved up to 50-fold improvement on the efficiency. The idea of using continuous flow is interesting. However, the presented results completely discarded the oxidation reactions accompanied with CO₂ reduction. The accompanied oxidation reaction, theoretically as water oxidation towards O₂ evolution, should be very difficult for the photocatalysts CdS, C₃N₄ and CuO. In most cases, self-oxidation should occur in the meantime of CO₂ reduction if O₂ evolution can not be observed. This self-oxidation issue would contradict with the claimed “long-term stability” even if the continuous flow is applied. Therefore, I could not recommend this manuscript for publication unless the authors can provide O₂ evolution results, which should be exactly correlated with quantity of CO₂ reduction product.

Response 1: We agree that the oxidative half-reaction should be explicitly identified and quantitatively coupled to the CO₂ reduction products. To address this concern, we performed comprehensive measurements of both O₂ and H₂O₂ evolution to determine the fate of photogenerated holes.

We minimized dissolved O₂ by thoroughly purging the aqueous phase with CO₂ prior to illumination. O₂ was analyzed by GC equipped with a thermal-conductivity detector (GC-TCD), while H₂O₂ was quantified by an iodometric UV-vis method⁴⁴. A calibration curve was prepared by recording the maximum absorbance at 350 nm for a series of H₂O₂ standards and fitting a straight line; time-series samples were then converted to concentration using this fit (spectra and calibration in **Supplementary Fig. 10**).

(New) **Supplementary Fig. 10 | Quantification of H₂O₂ by iodometry.** **a**, UV-vis absorption spectra of standard solutions with increasing H₂O₂ concentration⁵. **b**, Calibration curve at 350 nm (linear fit); H₂O₂ yields in reaction samples were determined from this line.

For TiO₂ in the *flow-enabled* reactor, we measured CO₂ reduction products together with the oxidative products over 4 h (**Supplementary Fig. 11**). H₂O₂ in the recirculating aqueous line was quantified by the uv-vis method and O₂ in the outlet gas by GC-TCD. Under identical conditions, H₂O₂ was the main oxygenated product, whereas O₂ was smaller. On a rate basis, the hole equivalents from H₂O₂ and O₂ (2- and 4-hole reactions, respectively) matched the electron

equivalents in CO and CH₄ (2- and 8-electron reactions, respectively) within experimental uncertainty ($e^-/h^+ \approx 1$), demonstrating quantitative coupling between the oxidative and reductive branches.

We performed the same analysis on g- C₃N₄ under visible light (**Supplementary Fig. 12**) as a representative material among those the Reviewer mentioned. At near-neutral pH, the oxidation of water to H₂O₂ (H₂O → H₂O₂) requires 1.76 V vs RHE (i.e., 1.346 V vs NHE)⁴⁵. Reported valence-band edges for g-C₃N₄ (+1.57 V vs NHE) and CdS (+1.86 V vs NHE) lie anodic to this level⁴⁶, indicating thermodynamic access to this pathway. Over a 1 h operation, O₂ did not exceed the GC-TCD detection limit, while H₂O₂ measured at levels sufficient to stoichiometrically account for the hole flux; the electron-hole balance was maintained through the two-electron pathway alone.

These observations are consistent with the band energetics and kinetics of these photocatalysts: two-hole water oxidation to H₂O₂ is thermodynamically feasible given the valence-band position and, on these materials, is kinetically more favorable than the four-hole oxygen evolution reaction under our *flow-enabled* system. Critically, the efficient consumption of photogenerated holes for productive H₂O₂/O₂ formation ($e^-/h^+ \approx 1$) indicates that holes are directed toward water oxidation rather than catalyst self-oxidation. This quantitative redox coupling demonstrates that our *flow-enabled* system successfully channels photogenerated charges into desired chemical transformations while preserving catalyst integrity. The measurements above therefore identify H₂O₂ as the prevailing oxidative product and provide the quantitative closure the reviewer requested.

(New) **Supplementary Fig. 11 | Comparison of O₂ Production with CO/CH₄ and the e^-/h^+ Ratio under TiO₂ photocatalytic CO₂ reduction under the *flow-enabled* system.** Conditions: TiO₂ 1 mg cm⁻²; CO₂ 1.2 bar, 2 sccm; H₂O 10 mL min⁻¹ (recirculating); Xe lamp 300 mW cm⁻². O₂ by GC-TCD; H₂O₂ by iodometric UV-vis (350 nm). Error bars: two independent experiment. Error bars in **a** represent standard deviations from three replicate experiments. All photocatalysts were tested under 300 mW·cm⁻² illumination with a Xe lamp.

(New) **Supplementary Fig. 12** | g-C₃N₄ under 1 hour visible light irradiation. production rates of H₂O₂ and CO (left) and the e⁻/h⁺ ratio (right). Conditions: g-C₃N₄ 1 mg cm⁻²; CO₂ 1.2 bar, 2 sccm; H₂O 10 mL min⁻¹. All photocatalysts were tested under 300 mW·cm⁻² illumination with a Xe lamp.

To address this concern, we have included new Supplementary Figures (**Supplementary Figs. 10–12**), added new references, and included a detailed explanation in the main text.

(*New Ref. 56*) Chen, Z., Yao, D., Chu, C. & Mao, S. Photocatalytic H₂O₂ production Systems: Design strategies and environmental applications. *Chem. Eng. J.* **451**, 138489 (2023).

(*New Ref. 57*) Sun, X. *et al.* Pairing oxygen reduction and water oxidation for dual-pathway H₂O₂ production. *Angew. Chem. Int. Ed Engl.* **63**, e202414417 (2024).

(*New Supplementary Ref. 5*) Ye, Y.-X. *et al.* Visible-light driven efficient overall H₂O₂ production on modified graphitic carbon nitride under ambient conditions. *Appl. Catal. B* **285**, 119726 (2021)

(*Main text*) To establish quantitative redox coupling, we measured oxidative products after CO₂ purging (O₂ measured by GC–TCD; H₂O₂ measured by iodometric UV–vis in **Supplementary Fig. 10**). Due to photocatalytic water oxidation reactions (WOR), a significant amount of H₂O₂ can be produced along with O₂, and the yield strongly depends on the reaction environment and the properties of the photocatalyst^{56,57}. As a result, hole equivalents (O₂, H₂O₂) matched electron equivalents in CO/CH₄ on TiO₂ (e⁻/h⁺ ≈ 1) and accounted for the hole flux on g-C₃N₄, confirming that photogenerated holes are efficiently consumed in productive water oxidation rather than catalyst degradation, thereby preserving catalyst integrity (**Supplementary Figs. 11–12**).

References.

1. Li, Y. *et al.* A modular tubular flow system with replaceable photocatalyst membranes for scalable coupling and hydrogenation. *Angew. Chem. Int. Ed Engl.* **62**, e202302979 (2023).
2. Yuan, Y. *et al.* Earth-abundant photocatalyst for H₂ generation from NH₃ with light-emitting diode illumination. *Science* **378**, 889–893 (2022).
3. Fang, S. *et al.* Photocatalytic CO₂ reduction. *Nat. Rev. Methods Primers* **3**, 1–21 (2023).
4. García de Arquer, F. P. *et al.* CO₂ electrolysis to multicarbon products at activities greater than 1 A cm⁻². *Science* **367**, 661–666 (2020).
5. Jung, H. *et al.* Continuous-flow reactor with superior production rate and stability for CO₂ reduction using semiconductor photocatalysts. *Energy Environ. Sci.* **16**, 2869–2878 (2023).
6. Albero, J., Peng, Y. & García, H. Photocatalytic CO₂ reduction to C₂+ products. *ACS Catal.* **10**, 5734–5749 (2020).
7. Cheng, S. *et al.* Emerging strategies for CO₂ photoreduction to CH₄: From experimental to data-driven design. *Adv. Energy Mater.* **12**, 2200389 (2022).
8. Wang, W. *et al.* Photocatalytic C-C coupling from carbon dioxide reduction on copper oxide with mixed-valence copper(I)/copper(II). *J. Am. Chem. Soc.* **143**, 2984–2993 (2021).
9. Yu, L. *et al.* Enhanced activity and stability of carbon-decorated cuprous oxide mesoporous nanorods for CO₂ reduction in artificial photosynthesis. *ACS Catal.* **6**, 6444–6454 (2016).
10. Wen, G. *et al.* Continuous CO₂ electrolysis using a CO₂ exsolution-induced flow cell. *Nat. Energy* **7**, 978–988 (2022).
11. Kim, B., Ma, S., Molly Jhong, H.-R. & Kenis, P. J. A. Influence of dilute feed and pH on electrochemical reduction of CO₂ to CO on Ag in a continuous flow electrolyzer. *Electrochim. Acta* **166**, 271–276 (2015).

12. Verma, S., Lu, X., Ma, S., Masel, R. I. & Kenis, P. J. A. The effect of electrolyte composition on the electroreduction of CO₂ to CO on Ag based gas diffusion electrodes. *Phys. Chem. Chem. Phys.* **18**, 7075–7084 (2016).
13. Xing, Z., Hu, L., Ripatti, D. S., Hu, X. & Feng, X. Enhancing carbon dioxide gas-diffusion electrolysis by creating a hydrophobic catalyst microenvironment. *Nat. Commun.* **12**, 136 (2021).
14. Simple levelized cost of energy (LCOE) calculator documentation. <https://www.nrel.gov/analysis/tech-lcoe-documentation?utm>.
15. Kumar, S. *et al.* P25@CoAl layered double hydroxide heterojunction nanocomposites for CO₂ photocatalytic reduction. *Appl. Catal. B* **209**, 394–404 (2017).
16. Wang, L. *et al.* Anchored Cu(II) tetra(4-carboxylphenyl)porphyrin to P25 (TiO₂) for efficient photocatalytic ability in CO₂ reduction. *Appl. Catal. B* **239**, 599–608 (2018).
17. Hwang, H. M. *et al.* Phase-selective disordered anatase/ordered rutile interface system for visible-light-driven, metal-free CO₂ reduction. *ACS Appl. Mater. Interfaces* **11**, 35693–35701 (2019).
18. Ye, M., Wang, X., Liu, E., Ye, J. & Wang, D. Boosting the photocatalytic activity of P25 for carbon dioxide reduction by using a surface-alkalinized titanium carbide MXene as cocatalyst. *ChemSusChem* **11**, 1606–1611 (2018).
19. Chen, Q. *et al.* Z-scheme Bi/AgBiS₂/P25 for enhanced CO₂ photoreduction to CH₄ and CO with photo-thermal synergy. *Appl. Surf. Sci.* **555**, 149648 (2021).
20. Xu, C. *et al.* Photothermal coupling factor achieving CO₂ reduction based on palladium-nanoparticle-loaded TiO₂. *ACS Catal.* **8**, 6582–6593 (2018).

21. Wang, Z. *et al.* Enhanced photocatalytic CO₂ reduction over TiO₂ using metalloporphyrin as the cocatalyst. *Catalysts* **10**, 654 (2020).
22. Jung, H. *et al.* Highly efficient and stable CO₂ reduction photocatalyst with a hierarchical structure of mesoporous TiO₂ on 3D graphene with few-layered MoS₂. *ACS Sustain. Chem. Eng.* **6**, 5718–5724 (2018).
23. Su, K. *et al.* In situ coating CsPbBr₃ nanocrystals with graphdiyne to boost the activity and stability of photocatalytic CO₂ reduction. *ACS Appl. Mater. Interfaces* **12**, 50464–50471 (2020).
24. Jiang, Z. *et al.* Living atomically dispersed Cu ultrathin TiO₂ nanosheet CO₂ reduction photocatalyst. *Adv. Sci. (Weinh.)* **6**, 1900289 (2019).
25. Lee, B.-H. *et al.* Electronic interaction between transition metal single-atoms and anatase TiO₂ boosts CO₂ photoreduction with H₂O. *Energy Environ. Sci.* **15**, 601–609 (2022).
26. He, Y., Wang, Y., Zhang, L., Teng, B. & Fan, M. High-efficiency conversion of CO₂ to fuel over ZnO/g-C₃N₄ photocatalyst. *Appl. Catal. B* **168–169**, 1–8 (2015).
27. Nie, N., Zhang, L., Fu, J., Cheng, B. & Yu, J. Self-assembled hierarchical direct Z-scheme g-C₃N₄/ZnO microspheres with enhanced photocatalytic CO₂ reduction performance. *Appl. Surf. Sci.* **441**, 12–22 (2018).
28. Zhang, F., Li, Y.-H., Qi, M.-Y., Tang, Z.-R. & Xu, Y.-J. Boosting the activity and stability of Ag-Cu₂O/ZnO nanorods for photocatalytic CO₂ reduction. *Appl. Catal. B* **268**, 118380 (2020).
29. Cao, Y. *et al.* Modulating electron density of vacancy site by single Au atom for effective CO₂ photoreduction. *Nat. Commun.* **12**, 1675 (2021).

30. Yu, J., Jin, J., Cheng, B. & Jaroniec, M. A noble metal-free reduced graphene oxide–CdS nanorod composite for the enhanced visible-light photocatalytic reduction of CO₂ to solar fuel. *J. Mater. Chem. A Mater. Energy Sustain.* **2**, 3407 (2014).
31. Wang, J. *et al.* A single Cu-center containing enzyme-mimic enabling full photosynthesis under CO₂ reduction. *ACS Nano* **14**, 8584–8593 (2020).
32. Ma, M. *et al.* Ultrahigh surface density of Co-N₂C single-atom-sites for boosting photocatalytic CO₂ reduction to methanol. *Appl. Catal. B* **300**, 120695 (2022).
33. Agarwal, V. G. & Haussener, S. Quantifying mass transport limitations in a microfluidic CO₂ electrolyzer with a gas diffusion cathode. *Commun. Chem.* **7**, 47 (2024).
34. Zhang, Z. *et al.* Cavity-confined Au@Cu₂O yolk-shell nanoreactors enable switchable CH₄/C₂H₄ selectivity. *Nat. Commun.* **16**, 7559 (2025).
35. Tan, Y. C., Lee, K. B., Song, H. & Oh, J. Modulating local CO₂ concentration as a general strategy for enhancing C–C coupling in CO₂ electroreduction. *Joule* **4**, 1104–1120 (2020).
36. Lum, Y. & Ager, J. W. Sequential catalysis controls selectivity in electrochemical CO₂ reduction on Cu. *Energy Environ. Sci.* **11**, 2935–2944 (2018).
37. Zhang, T. *et al.* Nickel–nitrogen–carbon molecular catalysts for high rate CO₂ electroreduction to CO: On the role of carbon substrate and reaction chemistry. *ACS Appl. Energy Mater.* **3**, 1617–1626 (2020).
38. Tan, Y. C. *et al.* Pitfalls and protocols: Evaluating catalysts for CO₂ reduction in electrolyzers based on gas diffusion electrodes. *ACS Energy Lett.* **7**, 2012–2023 (2022).
39. Kovalev, M. K., Ren, H., Zakir Muhamad, M., Ager, J. W. & Lapkin, A. A. Minor product polymerization causes failure of high-current CO₂-to-ethylene electrolyzers. *ACS Energy Lett.* **7**, 599–601 (2022).

40. Lees, E. W. *et al.* Linking gas diffusion electrode composition to CO₂ reduction in a flow cell. *J. Mater. Chem. A Mater. Energy Sustain.* **8**, 19493–19501 (2020).
41. Tazi, B. & Savadogo, O. Parameters of PEM fuel-cells based on new membranes fabricated from Nafion®, silicotungstic acid and thiophene. *Electrochim. Acta* **45**, 4329–4339 (2000).
42. Zhu, X. L. *et al.* Analysis by using X-ray photoelectron spectroscopy for polymethyl methacrylate and polytetrafluoroethylene etched by KrF excimer laser. *Appl. Surf. Sci.* **253**, 3122–3126 (2007).
43. Moore, T. *et al.* Electrolyzer energy dominates separation costs in state-of-the-art CO₂ electrolyzers: Implications for single-pass CO₂ utilization. *Joule* **7**, 782–796 (2023).
44. Ye, Y.-X. *et al.* Visible-light driven efficient overall H₂O₂ production on modified graphitic carbon nitride under ambient conditions. *Appl. Catal. B* **285**, 119726 (2021).
45. Guo, Y., Tong, X. & Yang, N. Photocatalytic and electrocatalytic generation of hydrogen peroxide: Principles, catalyst design and performance. *Nanomicro Lett.* **15**, 77 (2023).
46. Li, C. *et al.* Facile fabrication of CdS/Cu-doped g-C₃N₄ heterojunction for enhanced photocatalytic degradation of methylene blue. *J. Mater. Sci.: Mater. Electron.* **34**, (2023).

Responses to the Referees' Comments and the Corresponding Revisions

We extend our gratitude to the referees for their valuable feedback. After carefully considering their comments, we have made the 2nd revisions to the manuscript. We provide our responses to their comments below, along with the corresponding changes we have made to the main text and supplemental materials.

Response to Reviewer 4:

Comments¹. The proposed gas-liquid-solid three-phase interface construction strategy is not an original concept within the field. Numerous prior studies have demonstrated analogous three-phase interface electrodes in photocatalytic, photothermal catalytic, and photoelectrocatalytic applications (e.g., doi.org/10.1016/j.jcis.2025.01.187; doi.org/10.1016/j.ccllet.2025.112004; doi.org/10.1016/j.apsusc.2025.164650; doi.org/10.1038/s41467-025-60636-1). The role of such interfaces in enhancing mass transfer, enabling continuous flow operation, and improving catalytic performance has been extensively documented. Therefore, the manuscript must refine its scientific narrative to emphasize distinctive contributions that go beyond established knowledge.

Response 1: We thank the reviewer for this insightful comment, which has helped us better position our work within the broader context of three-phase interface research. Since our first report on the development of a continuous-flow photo-GDE reactor (Energy Environ. Sci., 16, 2869–2878 (2023)), we have further extended this platform to both photocatalytic and photoelectrocatalytic applications. **Because of its close structural and operational similarity to electrochemical GDE reactors, our photocatalytic flow system enables precise and reliable control of gas and liquid flow rates and pressures, which is critical for systematic investigation under realistic operating conditions.**

Most importantly, the significance of the present study is fundamentally different from that of prior reports on three-phase interfaces. **This work addresses one of the most critical challenges in photocatalytic CO₂ reduction: the severe operational instability of photocatalysts.** Despite decades of intensive research, most reported systems lose more than 80% of their initial activity within only a few hours, which has severely hindered practical and scalable implementation. Although incremental improvements in stability have been achieved through surface modification and cocatalyst optimization, these strategies remain insufficient for long-term, continuous operation. In this study, we demonstrate that **systematic control of CO₂ and H₂O flow conditions at the gas-liquid-solid interface leads to a record-level enhancement in operational stability, sustaining high production rates.** Even unmodified, common photocatalysts exhibit stable performance for over 100 hours, and continuous CO production is sustained for 15 days, representing, **to the best of our knowledge, the longest reported durability for photocatalytic CO₂ reduction to date. Notably, such a dramatic improvement in long-term stability solely through flow-condition engineering has not been reported previously.**

In the revised manuscript, we have cited and discussed recent representative studies on three-phase interface electrodes and clarified that the novelty of the present work does not lie in introducing the interface concept itself, but rather in uncovering the decisive role of *flow-enabled* three-phase interfaces in governing long-term stability in photocatalytic CO₂ reduction systems.

In the introduction of main text,

(Main text) From a reactor system-engineering perspective rather than further materials modification, many reported 'flow' photoreactors remain essentially two phase, most commonly liquid-gas or gas-solid, where the primary gains are improved illumination and phase-specific mass transport (Supplementary Fig. 1a-b). However, these configurations typically cannot maintain simultaneous and continuous contact of both CO₂ gas and H₂O, as the hydrogen source, at the catalyst surface. Consequently, for poorly soluble gases like CO₂, challenges such as product accumulation and catalyst flooding can persist. Numerous flow-based photoreactors, including gas-solid, liquid-solid, and gas-liquid-solid configurations, have been reported to enhance illumination and mass transport^{11,28-31} (Supplementary Fig. 1a-b). Nevertheless, most of these studies emphasize performance enhancement, while the role of continuous flow in regulating catalyst deactivation and long-term stability has not been systematically investigated.

Recognizing these limitations, our group recently developed a three phase (gas-liquid-solid) continuous flow photocatalytic system, inspired by gas diffusion electrode technology from electrocatalysis. A continuous-flow three-phase (gas-liquid-solid) photocatalytic system has recently been explored to improve reactant accessibility and mass transport³² (Supplementary Fig. 1c). This system demonstrated significantly enhanced production rates and stability by maintaining simultaneous gas and liquid flow at opposite sides of the catalyst layer. However, the fundamental mechanisms underlying this remarkable stability improvement remained unclear.

Also new refs were added in the introduction

(New Ref.30) Lin, X. et al. Monomer cylinder derived carbon nitride aerogel for flow photosynthesis of hydrogen peroxide. *Appl. Surf. Sci.* 716, 164650 (2026).

(New Ref.31) Wang, P. et al. Photothermal-electrocatalysis interface for fuel-cell grade ammonia harvesting from the environment. *Nat. Commun.* 16, 5581 (2025).

Comments 2. The CO₂ photocatalytic conversion efficiency reported in this work is not competitive when benchmarked against leading-edge systems. Moreover, critical performance metrics, such as the CO₂ conversion rate, are absent. To clearly establish the technical significance of this study, the authors are strongly advised to include a comprehensive comparison table incorporating recently published benchmark catalysts under comparable reaction conditions.

Response 2: We appreciate the reviewer's suggestion to contextualize our results within state-of-the-art systems. In response to the reviewer's suggestion, **we have now added a newly constructed state-of-the-art figure together with a performance summary table in the Supporting Information.** This comparison includes a broad range of recently reported photocatalytic CO₂ reduction systems, including those employing co-catalysts, heterostructures, and various hole scavengers, regardless of catalyst architecture or reaction medium. Using a widely used photocatalyst (TiO₂) and without introducing any structural or compositional modifications, we demonstrate that systematic control of gas and liquid flow conditions enables a substantial enhancement in long-term stability while maintaining a high and steady production rate. Notably, **the stability achieved at a comparable production rate is more than two orders of magnitude higher than that reported in previous studies (see the figure and table in the Supporting**

Information). Such simultaneous realization of sustained activity and prolonged durability, achieved solely through reactor-level and interfacial engineering, directly addresses one of the most critical bottlenecks in the field.

To address this comment, we added new **Supplementary Fig.4** and **Supplementary Table 2** referred it in main text.

(new) **Supplementary Fig. 4** | Comparison of state-of-the-art photocatalytic CO₂ reduction systems. Performance metrics and reaction durations of recently reported photocatalysts, encompassing various catalyst architectures (e.g., co-catalysts, heterostructures) and reaction conditions (e.g., hole scavengers).

(new) **Supplementary Table 2** | Performance comparison with state-of-the-art photocatalytic CO₂ reduction systems under various catalyst designs and reaction conditions.

Photocatalyst ^a	Reaction Medium	Light Source	Production Rate ^b (μmol/g·hr)	Stability ^c (hr)	Reaction system ^d	Ref.
TiO ₂ (P25)	CO ₂ and H ₂ O flow	300W Xe Lamp	686 (CO: 686)	360	Continuous flow reactor	This work
CuAu-DAs-TiO ₂	CO ₂ and H ₂ O	300W Xe Lamp	483.2 (C ₂ H ₄ : 568.8)	120	Intermittent flow system	5
Co ₃ O ₄ -TiO ₂ /CdS-CuO _x	CO ₂ gas with water drop	300 W Xe lamp	126 (CO: 124 CH ₄ : 2.5)	20	Gas Phase Batch	6
GDY-Cu/ZnO	CO ₂ gas with water drop	300W Xe Lamp	35.79 (CO: 2.12 CH ₄ : 33.67)	40	Gas Phase Batch	7
Pd ₁ -Bi ₁ Bi ₂ O ₃ /TiO ₂	CO ₂ gas with water drop	300W Xe Lamp	212.5 (C ₂ H ₆ : 212.5)	50	Gas Phase Batch	8

Br-COF @ BiOCl	CO ₂ gas from NaHCO ₃ +H ₂ SO ₄	300W Xe Lamp with 320nm filter	27.4 (CO: 27.4)	20	Gas phase batch	⁹
Cs ₂ AgBiBr ₆ @Co ₃ O ₄	CO ₂ gas with water drop	300W Xe Lamp with visible filter (> 420nm)	~200 (CO: ~200)	10	Gas phase batch	¹⁰
Co-doped CdS	CO ₂ purged solution (H ₂ O+CH ₃ CN+TEA)	300W Xe Lamp	2322.8 (CO: ~2322.8)	20	Liquid phase batch	¹¹
Cu ₂ O-Pt/SiC/IrO _x	CO ₂ bubbled to solution (2 mM FeCl ₂)	300W Xe Lamp with visible filter (> 420nm)	896.7 (HCOOH: 896.7)	40	Liquid phase batch	¹²
^a Photocatalysts for CO ₂ photoreduction. ^b Production rate calculated based on the amount of total products formed during CO ₂ reduction reaction, with values shown in parentheses. ^d Test time for photocatalytic durability including total cycle time. ^e Reaction system, either a static pressurized gas system or CO ₂ -purged water based liquid system.						

Also new refs were added in Supplementary Information

(New Ref. 5) Xie, Z. et al. Well-defined diatomic catalysis for photosynthesis of C₂H₄ from CO₂. Nat. Commun. 15, 2422 (2024).

(New Ref. 6) Lin, K. et al. Hollow TiO₂/CdS Z-Scheme Heterojunctions with Spatially Separated Cocatalysts for Highly Selective Photodriven CO₂ Conversion. Adv. Funct. Mater. e15276 (2025)

(New Ref. 7) Han, Y. et al. Slow-light-driven photocatalytic CO₂ reduction to CH₄ mediated by photonic crystal Graphdiyne-Cu/ZnO Z-scheme system. Chem. Eng. J. 500, 157636 (2024).

(New Ref. 8) Chen, Q. et al. Enhancing local CO₂ availability via amorphous Bi₂O₃ enable efficient and stable photocatalytic C₂H₆ production. Chem. Eng. J. 504, 158854 (2025).

(New Ref. 9) Wang, Y. et al. N-Bi covalently connected Z-scheme heterojunction by in situ anchoring BiOCl on triazine-based bromine-substituted covalent organic frameworks for the enhanced photocatalytic reduction of CO₂ and Cr (VI). Chem. Eng. J. 505, 159349 (2025).

(New Ref. 10) Song, Y. et al. Boosted photocatalytic CO₂ conversion of a Cs₂AgBiBr₆@Co₃O₄ composite with high activity and selectivity under low-concentration CO₂ and natural sunlight. Appl. Catal. B 363, 124816 (2025).

(New Ref. 11) Xiong, R. et al. Dual electronic-spin engineering in cobalt-doped CdS: Photocatalytic pathway switching from competing H₂ evolution to selective CO₂ reduction. Chem. Eng. J. 520, 166224 (2025).

(New Ref. 12) Wang, Y. et al. Direct and indirect Z-scheme heterostructure-coupled photosystem enabling cooperation of CO₂ reduction and H₂O oxidation. Nat. Commun. 11, 3043 (2020).

(Main text) Remarkably, the *flow-enabled* platform demonstrated significantly enhanced stability. For instance, TiO₂ retained over 80% of its initial activity for more than 360 hours (>15 days), achieving cumulative product formation rates up to 825 μmol·g⁻¹·h⁻¹—far exceeding the performance observed under batch conditions without CO₂ and H₂O flow. This catalyst exhibits remarkable stability, surpassing that of recently reported semiconductor-based CO₂ photocatalysts incorporating co-catalysts or heterostructures. The carbon source of CO production in this *flow-enabled* reactor configuration has been verified through ¹³CO₂ isotope labeling experiments in our previous work ¹, confirming that CO originates from photocatalytic CO₂ reduction. This represents

a more than two order of magnitude enhancement in stability compared to previously reported TiO₂-based photocatalysts²⁻¹⁹, and even outperforms recent systems employing co-catalysts or heterostructure engineering (Supplementary Fig.4 & Supplementary Table 2), demonstrating showing the potential of flow-regulated operation in overcoming longstanding barriers to practical photocatalytic CO₂ reduction.

Comments 3. The authors' attempt to draw parallels between electrochemical gas diffusion electrodes and the flow-enabled photocatalytic system to explain enhanced mass transfer is conceptually suggestive but lacks rigor. Although structural similarities exist in device design, the operational mechanisms differ significantly due to distinct driving forces and reaction environments. The innovative application of analytical titration principles to quantitatively assess the theoretical CO₂ mass transfer efficiency across different reactor configurations represents a methodologically sound advancement. This approach provides indirect evidence that the engineered three-phase interface substantially increases gas-liquid contact area compared to conventional configurations, thereby facilitating interfacial mass transfer. This methodological contribution holds value and broader implications for reactor engineering. However, to prevent misinterpretation, the authors should explicitly label the time point of each image in Figure S3.

Response 3: We thank the reviewer for the constructive clarification regarding the analogy with electrochemical gas diffusion electrodes. As the reviewer suggested, in the revised manuscript, we avoid any implication of identical transport mechanisms and instead frame the discussion in terms of qualitative mass-transfer considerations arising from enhanced gas-liquid contact under continuous-flow conditions, without claiming direct equivalence to electrochemical GDE operation.

In the main text,

(Main text) ~~Recognizing these limitations, our group recently developed a three phase (gas-liquid-solid) continuous-flow photocatalytic system, inspired by gas diffusion electrode technology from electrocatalysis three-phase (gas-liquid-solid) photocatalytic systems have recently been explored to improve reactant accessibility and mass transport³² (Supplementary Fig. 1c).~~

(Main text) It is likely that H₂O flow has a greater influence on the initial reaction activity, possibly due to infiltration of H₂O into the gas diffusion ~~electrode (GDE) layer~~ under no-flow condition, which adversely affect the formation and maintenance of the triple-phase interface^{1,20,21}.

Importantly, the quantitative assessment of CO₂ mass transfer in our work is based solely on reactor-intrinsic CO₂ uptake measurements, independent of electrochemical models. To prevent any potential misinterpretation, we have also explicitly labeled the time points for each image in **Supplementary Fig. 3**.

(Revised) **Supplementary Fig. 3 | Time-lapse images of the phenolphthalein decolorization during CO₂ exposure for the same geometric gas-liquid contact area.** **a**, The *flow-enabled* cell decolorizes uniformly across the interface. Inset: 0.10 M NaOH solution before and after the test. **b**, The *flow-less* (pressurized gas line in headspace of bottle) cell decolorizes from the gas-liquid interface downward, leaving the bottom region colored longest (yellow arrow), consistent with non-uniform CO₂ delivery and slower intra-liquid mixing in the batch geometry. Contact area = 1 cm diameter $\approx 0.785 \text{ cm}^2$, without light irradiation and photocatalyst.

Comments 4. In response to Reviewer 1's 4th comment, the authors have not provided a direct or satisfactory explanation. To substantiate their claims regarding flow-dependent performance, experimental data on CO₂ conversion rates under systematically varied gas flow rates must be included. Such data are essential for establishing a causal relationship between hydrodynamic conditions and catalytic output.

Response 4: We appreciate the reviewer's comment regarding the flow-dependent catalytic performance. In **Supplementary Fig. 8**, we demonstrated that the presence and combination of CO₂ and H₂O flows critically govern catalytic stability and surface deactivation behavior. We also agree that further clarification on the flow-rate-dependent conversion is needed. Accordingly, we have expanded the discussion for relationship between hydrodynamic conditions and catalytic output in **Supplementary Fig. 8**:

Supplementary Fig. 8 | Effects of various factors on the flow-type reactor system. a, Effect of applied CO_2 gas pressure. **b**, Effect of gas flow rate on the production rate. **c**, Effects of the cycled electrolyte flow rate on the production rates of CO and CH_4

In our *flow-enabled* system, the formation and stability of the three-phase interface are governed by the interplay between electrolyte flow rate, gas flow rate, and gas pressure. Each parameter has an optimal value: below this threshold, mass transport^{1,22} becomes inefficient due to insufficient reactant supply or product removal; above it, the three-phase interface is disrupted—either by electrolyte penetration into the catalyst layer or by reduced residence time that limits effective adsorption and reaction. This behavior is consistent with a transition from a mass-transfer-limited regime to a steady-state regime, confirming that hydrodynamic conditions directly govern catalytic performance.

Comments 5. The discussion on the microscopic mechanism underlying the synergistic interaction between CO₂ and H₂O at the three-phase interface remains insufficient. Beyond macroscopic performance comparisons, the authors should provide mechanistic insights into microscale transport phenomena. Specifically, analysis of CO₂ and H₂O diffusion behavior and adsorption kinetics in porous media under different gas-liquid flow rates, supported by in-situ spectrum, electron microscope image or computational simulations, would strengthen the interpretation of dynamic interfacial processes.

Response 5: We sincerely thank the reviewer for the insightful comments. We fully agree that microscopic investigation of interfacial processes is highly important, and we also attempted to elucidate the behavior of CO₂ and H₂O molecules on the photocatalyst surface using in-situ spectroscopic techniques. Specifically, we explored both in-situ FT-IR and in-situ Raman spectroscopy to directly probe molecular dynamics at the gas-liquid-solid interface.

For in-situ FT-IR measurements, we designed and fabricated a custom three-phase flow cell. However, the complexity of the cell geometry and its incompatibility with the optical configuration of the FT-IR instrument prevented reliable signal acquisition. For in-situ Raman spectroscopy, although a suitable flow-cell configuration was successfully constructed, a fundamental experimental challenge arose from the interference between the Raman excitation laser and the ultraviolet light required for photocatalyst activation. In the three-phase flow configuration, stronger UV irradiation is required than in conventional batch photocatalytic systems due to light attenuation by the liquid layer. As a result, the intense UV background overwhelmed the Raman signals, precluding meaningful spectral analysis. Due to these technical limitations, we were unable to obtain reliable in-situ spectroscopic evidence within the scope of the present study. We therefore state that detailed in-situ interfacial characterization will be the subject of future work.

Instead, we performed **molecular dynamics (MD) simulations** to investigate the molecular behavior at the gas-liquid-solid interface under flow conditions (**Figure R1**). The simulations were carried out in a periodically replicated simulation box with a maximum dimension of ~10 nm, containing 35,000 H₂O molecules and 200 CO₂ molecules, and an effective reaction distance of 4 Å from the TiO₂ surface. An external pressure and a driving force were applied along the z-axis to mimic gas flow (Figure R1a). It should be noted that electron transfer, applied potential, and chemical reactions were not explicitly considered in this MD framework.

The simulation results provide mechanistic insight into the role of flow in governing interfacial molecular distributions. Under liquid-phase *flow-less* conditions, H₂O molecules preferentially adsorb on the TiO₂ surface, thereby hindering the access of CO₂ and reducing the probability of interfacial reactions. Even under gas-phase *flow-less* conditions in a humidified CO₂ environment (800 H₂O and 200 CO₂ molecules), H₂O molecules are found to dominate surface adsorption when both species are present. In contrast, under forced-flow conditions mimicking a *flow-enabled*

system, the average number of CO₂ molecules within the effective reaction distance ($< 4\text{Å}$) increases systematically with increasing external driving force, and the flow-assisted system exhibits the highest average coordination number within 4Å of the TiO₂ surface. These results indicate that forced flow promotes the transport of CO₂ molecules into the reactive interfacial zone, thereby increasing their residence probability near active sites and, consequently, the likelihood of photocatalytic reactions. **We intend to report these MD results together with comprehensive in-situ microscopic characterization in a future study.**

Figure S1. Molecular dynamic simulation of CO₂ and H₂O behavior on TiO₂ photocatalyst. **a**, Schematic of the simulation system of TiO₂ particles on PTFE membrane (left) and applying H₂O and CO₂ molecules (right). **b**, Average number of CO₂ molecules (CN) as a function of distance from TiO₂ surface under *flow-enabled* and *flow-less* conditions in liquid and gas phases. **c**, Enlarged view at an effective reaction distance of 4Å .

Comments 6. Regarding Reviewer 1's 9th comment, observed differences in the Cu K-edge extended X-ray absorption fine structure (EXAFS) spectra between the flow-enabled and flow-less systems suggest potential structural distinction. The authors must conduct a rigorous assessment to determine whether these structural distinctions influence catalytic activity. It is imperative to strictly decouple material-level structural distinction from hydrodynamic effects to ensure that performance differences are only attributable to flow-induced mass transfer enhancement.

Response 6: We thank the reviewer for raising the possibility of structural contributions suggested by the Cu K-edge EXAFS results. As demonstrated in **Supplementary Fig. 23**, the in-situ XAFS measurements were performed using the same CuO catalyst with identical chemical composition and initial coordination environment in both the *flow-enabled* and *flow-less* configurations.

To address the reviewer's concern regarding decoupling structural and hydrodynamic effects, we highlight two key observations: First, within the first hour of photocatalytic reaction, structural change of catalyst could not be observed at both *flow-less* and *flow-enabled* system. Nevertheless, the *flow-enabled* system already exhibited notably higher catalytic activity compared to the *flow-less* system. This indicates that **the performance gap at this early stage arises from flow-regulated mass transfer rather than material-level structural differences**. Second, during prolonged operation (5hr reaction), local coordination changes of the catalyst could be observed under *flow-less* condition, which is driven by the accumulation of reaction intermediates altering the local bonding environment around Cu (**Fig. 4**). **While some degree of structural change cannot be ruled out as a contributing factor to deactivation in the *flow-less* system, the dominant cause of deactivation should be carbon accumulation** observed exclusively under *flow-less* conditions, even when the catalyst structure remains stable (**Fig. 3**).

Accordingly, we have revised **Supplementary Fig. 24** to better clarify the relationship between structural evolution, carbon accumulation, and catalytic performance under each condition.

(Revised) Supplementary Fig. 24 | Performance of CuO under *flow-enabled* and batch operation. a, Stability over a 100 h reaction. b, Production rate of CuO in *flow-enabled* system and c, in *flow-less* system at 1, 2, 5, 25 h. Reaction conditions: Xe lamp 100 mW cm^{-2} ; CO_2 10 sccm; H_2O 10 mL min^{-1}

Comments 7. With respect to Reviewer 1's 10th comment, the presence of a C-C signal in XPS is commonly attributed to adventitious carbon contamination (amorphous carbon) originating from ambient exposure, which serves as the "internal standard" for peak position calibration and correction of all other elements. The authors' explanation is scientifically valid and fully consistent with standard practices in materials characterization.

Response 7: We sincerely thank the reviewer for the positive evaluation of our explanation regarding the C-C signal in the XPS spectra. We are pleased that this interpretation is fully consistent with standard practices in materials characterization.

Responses to the Referees' Comments and the Corresponding Revisions

We sincerely thank for the positive evaluation of our manuscript and for the constructive suggestions to further improve clarity. We have carefully revised the manuscript accordingly. Our detailed responses are provided below.

Response to Reviewer 4:

Comments 1. To enhance clarity for readers unfamiliar with the experimental setup, the authors should explicitly clarify the relationship among the X-axis variables across the three subplots in Figure S8, and explicitly state whether the reported pressure and flow rate values correspond to identical conditions on both sides of the catalytic membrane.

Response 1: We appreciate this important suggestion. To eliminate ambiguity, we have added a new schematic (**Supplementary Fig. 8a**) that illustrates the flow configuration on both sides of the photocatalyst layer. The CO₂ gas pressure and gas flow rate are applied on the gas-diffusion side of the catalytic membrane, while the water flow rate corresponds to the liquid side. These parameters were independently controlled on each side of the membrane. The figure caption and the corresponding text in the supplementary materials have been revised accordingly as follows:

(Revised) **Supplementary Fig. 8 | Effects of various factors on the flow-enabled system. a,** Schematic illustration of gas and water flows inside the reactor. **b,** Effect of applied CO₂ gas

pressure on the production rate. **c**, Effect of gas flow rate on the production rate. **d**, Effects of the cycled water flow rate on the production rates. Experiments in (b–d) were conducted under $300 \text{ mW}\cdot\text{cm}^{-2}$ irradiation from a 300 W Xe lamp, with error bars representing standard deviations from at least three identical experiments.

(Supplementary text) In our flow-enabled system, the formation and stability of the three-phase interface are governed by the interplay between water flow rate, gas flow rate, and gas pressure, each of which can be independently controlled. The CO_2 gas pressure and gas flow rate are applied on the gas-diffusion side of the catalytic membrane, while the water flow rate corresponds to the liquid side (**Supplementary Fig.8a**). Each parameter has an optimal value: below this threshold, mass transport becomes inefficient due to insufficient reactant supply or product removal; above it, the three-phase interface is disrupted—either by electrolyte penetration into the catalyst layer or by reduced residence time that limits effective adsorption and reaction^{13,14}. This behavior is consistent with a transition from a mass-transfer-limited regime to a steady-state regime, confirming that hydrodynamic conditions directly govern catalytic performance.

Comments 2. In Figure S1a, the color-coded particles represent distinct chemical species; the legend and caption must unambiguously identify CO_2 , H_2O , and TiO_2 . Additionally, Figure S1c is currently missing a legend and must be supplemented accordingly.

Response 2: We thank the Editor for pointing out this clarity issue. We have revised both the schematic and the caption to explicitly define each color-coded species as follows:

(Revised) Supplementary Fig. 1 | Photocatalytic reactor system configuration. a, Schematic illustrations of flow-less two-phase batch reactors; **b**, batch-modified flow reactors; and **c**, a gas diffusion electrode (GDE)-modified three-phase flow reactor. (left) Schematic illustration of the continuous flow-enabled photocatalytic reactor showing key components: gas pressure regulator, gas flow plate, mass flow meter (MFM), peristaltic pump, and PTFE membrane with photocatalyst coating. (right) Detailed gas diffused photocatalyst layer highlighting the role of the triple-phase interface in facilitating mass transport of CO₂ and H₂O. Color-coded particles represent C (gray), O (red), H (light blue), and TiO₂ photocatalyst (blue).